# DISTILLING ODE SOLVERS OF DIFFUSION MODELS INTO SMALLER STEPS

## ABSTRACT

Distillation techniques have substantially improved the sampling speed of diffusion models, allowing of the generation within only one step or a few steps. However, these distillation methods require extensive training for each dataset, sampler, and network, which limits their practical applicability. To address this limitation, we propose a straightforward distillation approach, *Distilled-ODE solvers* (D-ODE solvers), that optimizes the ODE solver rather than training the denoising network. D-ODE solvers are formulated by simply applying a single parameter adjustment to existing ODE solvers. Subsequently, D-ODE solvers with smaller steps are optimized by ODE solvers with larger steps through distillation over a batch of samples. Our comprehensive experiments indicate that D-ODE solvers outperform existing ODE solvers, including DDIM, PNDM, DPM-Solver, DEIS, and EDM, especially when generating samples with fewer steps. Our method incur negligible computational overhead compared to previous distillation techniques, enabling simple and rapid integration with previous samplers. Qualitative analysis further shows that D-ODE solvers enhance image quality while preserving the sampling trajectory of ODE solvers.

## 1 INTRODUCTION

Diffusion models (Sohl-Dickstein et al., 2015; Ho et al., 2020; Song & Ermon, 2019) have recently gained attention as a promising framework for generative models, demonstrating state-of-the-art performance across a wide range of applications. These models are designed to progressively remove noise from a sample during training and generate new data samples from a predefined prior distribution during inference. They have achieved notable success in various domains, including image generation (Song et al., 2020b; Dhariwal & Nichol, 2021), text generation (Hoogeboom et al., 2021; Austin et al., 2021), audio generation (Mittal et al., 2021; Lu et al., 2021), 3D shape generation (Cai et al., 2020; Luo & Hu, 2021), video synthesis (Harvey et al., 2022; Yang et al., 2022b), and graph generation (Niu et al., 2020; Vignac et al., 2023).

Despite their ability to produce high-quality samples and mitigate issues such as mode collapse (Salimans et al., 2016; Zhao et al., 2018), the sampling process of diffusion models typically involves a substantial number of network evaluations, rendering the process slow and computationally intensive (Xiao et al., 2021). Consequently, recent research has concentrated on accelerating or optimizing the sampling process while preserving the quality of generated samples (Song et al., 2020a; Karras et al., 2022; Salimans & Ho, 2021). In particular, methods aimed at improving the sampling efficiency of diffusion models can be broadly categorized into two groups: *learning-free sampling* and *learning-based sampling* (Yang et al., 2022a).

Learning-free sampling can be applied to pre-trained diffusion models without additional training and typically relies on efficient solvers for stochastic differential equations (SDEs) or ordinary differential equations (ODEs) (Song et al., 2020b). For instance, DDIM (Song et al., 2020a) employs a non-Markovian process to accelerate sampling. PNDM (Liu et al., 2021) introduces a pseudo-numerical method for solving differential equations on given data manifolds. EDM (Karras et al., 2022) utilizes Heun's second-order method and demonstrates improved sampling quality compared to the naive Euler's method (Song et al., 2020b). More recently, methods like DPM-Solver (Lu et al., 2022) and DEIS (Zhang & Chen, 2022) leverage the semi-linear structure of diffusion ODEs and employ numerical methods of exponential integrators.

On the other hand, learning-based sampling requires additional training to optimize specific learning objectives, such as knowledge distillation (Salimans & Ho, 2021; Song et al., 2023) and optimized discretization (Nichol & Dhariwal, 2021; Watson et al., 2021). For example, progressive distillation (Salimans & Ho, 2021) iteratively distills pre-trained diffusion models into a student model that requires only half the number of sampling steps. Recently, Song et al. (2023) introduces consistency models, which are trained to predict consistent outputs along the same ODE trajectory. Consistency models can be trained independently or with knowledge distillation.

While learning-free and learning-based sampling have been studied independently, their combination remains relatively unexplored. In this paper, we propose a novel distillation method for diffusion models, Distilled-ODE solvers (D-ODE solvers), that leverages the underlying principles of existing ODE solvers. Our approach is grounded in a key observation that the outputs of denoising networks exhibit high correlation within neighboring time steps. D-ODE solvers introduce a single additional parameter to ODE solvers, optimized by minimizing the difference between the output of D-ODE solvers with smaller steps (student) and that of ODE solvers with larger steps (teacher). Once the optimal parameter for D-ODE solvers is established, it can be reused across different batches during sampling, while keeping the denoising network fixed. Our method represents an intersection of learning-free and learning-based sampling, employing a straightforward distillation process to optimize D-ODE solvers while capitalizing on the sampling dynamics inherent in ODE solvers.

Our main contributions can be summarized as follows:

- We introduce Distilled-ODE solvers (D-ODE solvers), which transfer the knowledge from ODE solvers with larger steps to those with smaller steps through a simple formulation.
- D-ODE solvers significantly reduce distillation times by optimizing existing ODE solvers and eliminate the need for extensive parameter updates in pre-trained denoising networks.
- In quantitative studies, our new sampler outperforms state-of-the-art ODE solvers in terms of FID scores on several image generation benchmarks.

## 2 BACKGROUND

**Forward and reverse diffusion processes:** The forward process $\{\boldsymbol{x}_t \in \mathbb{R}^D\}_{t \in [0,T]}$ initiates with $\boldsymbol{x}_0$ drawn from the data distribution $p_{data}(\boldsymbol{x})$ and evolves to $\boldsymbol{x}_T$ at timestep $T > 0$. Given $\boldsymbol{x}_0$, the distribution of $\boldsymbol{x}_t$ can be expressed as follows:

$$q_t(\boldsymbol{x}_t|\boldsymbol{x}_0) = \mathcal{N}(\boldsymbol{x}_t|\alpha_t \boldsymbol{x}_0, \sigma_t^2 \boldsymbol{I}), \tag{1}$$

where $\alpha_t \in \mathbb{R}$ and $\sigma_t \in \mathbb{R}$ determine the noise schedule of the diffusion models, with the signal-to-noise ratio (SNR) $\alpha_t^2/\sigma_t^2$ strictly decreasing as $t$ progresses (Kingma et al., 2021). This ensures that $q_T(\boldsymbol{x}_T)$, the distribution of $\boldsymbol{x}_T$, approximates pure Gaussian noise in practice.

The reverse process of diffusion models is approximated using a denoising network to iteratively remove noise. Starting from $\boldsymbol{x}_T$, the reverse process is defined with the following transition (Ho et al., 2020):

$$p_\theta(\boldsymbol{x}_{t-1}|\boldsymbol{x}_t) = \mathcal{N}(\boldsymbol{x}_{t-1}|\mu_\theta(\boldsymbol{x}_t, t), \Sigma_\theta(\boldsymbol{x}_t, t)), \tag{2}$$

where $\theta$ represents the trainable parameters in the denoising network, and $\mu_\theta(\boldsymbol{x}_t, t)$ and $\Sigma_\theta(\boldsymbol{x}_t, t)$ are the Gaussian mean and variance estimated by the network $\theta$.

**SDE and ODE formulation:** Song et al. (2020b) formulate the forward diffusion process using a stochastic differential equation (SDE) to achieve the same transition distribution as Equation 1:

$$d\boldsymbol{x}_t = f(t)\boldsymbol{x}_t dt + g(t)d\boldsymbol{w}_t, \quad \boldsymbol{x}_0 \sim p_{data}(\boldsymbol{x}), \tag{3}$$

where $\boldsymbol{w}_t \in \mathbb{R}^D$ is the standard Wiener process, and $f(t)$ and $g(t)$ are functions of $\alpha_t$ and $\sigma_t$. Song et al. (2020b) also introduce the reverse-time SDE, which evolves from timestep $T$ to 0, based on Anderson (1982):

$$d\boldsymbol{x}_t = [f(t)\boldsymbol{x}_t - g^2(t)\nabla_{\boldsymbol{x}} \log q_t(\boldsymbol{x}_t)]dt + g(t)d\bar{\boldsymbol{w}}_t, \quad \boldsymbol{x}_T \sim q_T(\boldsymbol{x}_T), \tag{4}$$

where $\bar{\boldsymbol{w}}_t$ is the standard Wiener process in reverse time, and $\nabla_{\boldsymbol{x}} \log q_t(\boldsymbol{x}_t)$ is referred to as the score function (Hyvärinen & Dayan, 2005). The randomness introduced by the Wiener process can

be omitted to define the diffusion ordinary differential equation (ODE) in the reverse process, which corresponds to solving the SDE on average:

$$d\boldsymbol{x}_t = [f(t)\boldsymbol{x}_t - \frac{1}{2}g^2(t)\nabla_{\boldsymbol{x}}\log q_t(\boldsymbol{x}_t)]dt \quad \boldsymbol{x}_T \sim q_T(\boldsymbol{x}_T). \tag{5}$$

The formulation of the probability flow ODE opens up possibilities for using various ODE solvers to expedite diffusion-based sampling processes (Liu et al., 2021; Lu et al., 2022; Zhang & Chen, 2022; Karras et al., 2022).

**Denoising score matching:** To solve Equation 5 during sampling, the score function $\nabla_{\boldsymbol{x}}\log q_t(\boldsymbol{x}_t)$ must be estimated. Ho et al. (2020) propose estimating the score function using a noise prediction network $\epsilon_{\boldsymbol{\theta}}$ such that $\nabla_{\boldsymbol{x}}\log q_t(\boldsymbol{x}_t) = -\epsilon_{\boldsymbol{\theta}}(\boldsymbol{x}_t, t)/\sigma_t$ with $\boldsymbol{x}_t = \alpha_t\boldsymbol{x} + \sigma_t\epsilon$. The noise prediction network $\epsilon_{\boldsymbol{\theta}}$ is trained using the $L_2$ norm, given samples drawn from $p_{data}$:

$$\mathbb{E}_{\boldsymbol{x}\sim p_{data}}\mathbb{E}_{\boldsymbol{\epsilon}\sim\mathcal{N}(\boldsymbol{0},\sigma_t^2\boldsymbol{I})}||\epsilon_{\boldsymbol{\theta}}(\alpha_t\boldsymbol{x} + \sigma_t\boldsymbol{\epsilon}, t) - \boldsymbol{\epsilon}||^2. \tag{6}$$

Here, Gaussian noise is added to the data $\boldsymbol{x}$ following the noise schedule $(\alpha_t, \sigma_t)$, and the noise prediction network predicts the added noise $\boldsymbol{\epsilon}$ from the noisy sample.

Alternatively, the score function can be represented using a data prediction network $x_{\boldsymbol{\theta}}$ instead of $\epsilon_{\boldsymbol{\theta}}$ with $\nabla_{\boldsymbol{x}}\log q_t(\boldsymbol{x}_t) = (x_{\boldsymbol{\theta}}(\boldsymbol{x}_t, t) - \boldsymbol{x}_t)/\sigma_t^2$. The data prediction network $x_{\boldsymbol{\theta}}$ is trained with following $L_2$ norm:

$$\mathbb{E}_{\boldsymbol{x}\sim p_{data}}\mathbb{E}_{\boldsymbol{\epsilon}\sim\mathcal{N}(\boldsymbol{0},\sigma_t^2\boldsymbol{I})}||x_{\boldsymbol{\theta}}(\alpha_t\boldsymbol{x} + \sigma_t\boldsymbol{\epsilon}, t) - \boldsymbol{x}||^2. \tag{7}$$

It is worth noting that estimating the original data $\boldsymbol{x}$ is theoretically equivalent to learning to predict the noise $\boldsymbol{\epsilon}$ (Ho et al., 2020; Luo, 2022). While some works argue that predicting the noise empirically results in higher quality samples (Ho et al., 2020; Saharia et al., 2022), Karras et al. (2022) recently achieved state-of-the-art performance using the data prediction network. In this work, we conduct comprehensive experiments with both noise and data prediction networks.

## 3 METHOD

As introduced in Section 1, our study aims to bridges the gap between learning-based and learning-free sampling, leveraging the advantages of both approaches. We utilize the sampling dynamics of ODE solvers while enhancing sample quality through a simple and efficient distillation process. This section commences with a fundamental observation of the high correlation among denoising outputs, which motivates the formulation of D-ODE solvers. We then delve into the details of transferring knowledge from ODE solvers to D-ODE solvers.

### 3.1 CORRELATION BETWEEN DENOISING OUTPUTS

ODE solvers typically improve the sampling process by exploiting the denoising network's output history, allowing for the omission of many intermediate steps. Hence, comprehending the connections between denoising outputs is paramount when developing D-ODE Solvers. Our aim is to create novel ODE solvers that harness the benefits of sampling dynamics while keeping optimization degrees of freedom to a minimum.

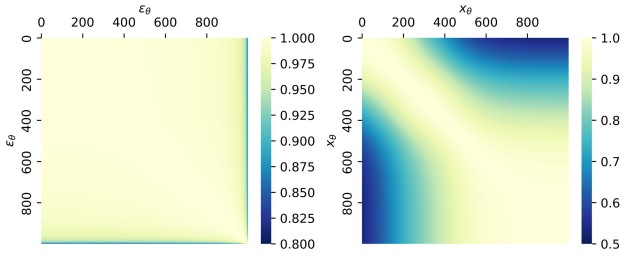

Figure 1: Heatmaps are drawn by cosine similarity among denoising outputs with 1000-step DDIM on CIFAR-10. Noise prediction model (left) and data prediction model (right).

Figure 1 presents heatmaps based on cosine similarity calculations between all denoising outputs from a 1000-step DDIM (Song et al., 2020a) run. We observe that predictions from neighboring timesteps exhibit high correlations in both denoising networks, with cosine similarities close to one. This observation suggests that denoising outputs contain redundant and duplicated information, allowing us to skip the evaluation of denoising networks for most

timesteps. For example, we can combine the history of denoising outputs to better represent the next output, effectively reducing the number of steps required for accurate sampling. This idea is implemented in most ODE solvers, which are formulated based on the theoretical principles of solving differential equations. These solvers often adopt linear combinations or multi-step approaches, leveraging previous denoising outputs to precisely estimate the current prediction (Watson et al., 2021; Liu et al., 2021; Karras et al., 2022; Lu et al., 2022; Zhang & Chen, 2022). Consequently, ODE solvers can generate high-quality samples with far fewer timesteps compared to the thousand steps of DDPM (Ho et al., 2020).

## 3.2 FORMULATION OF D-ODE SOLVER

We now introduce a D-ODE solver with a straightforward parameterization to distill knowledge from ODE solvers. We begin by outlining a fundamental method for representing the new denoising prediction $\tilde{D}_t$ at timestep $t$ as a linear combination of current and previous denoising outputs $\{D_{\boldsymbol{\theta}}(\hat{\boldsymbol{x}}_k, k)\}_{k=t}^{T}$:

$$\tilde{D}_t = \sum_{k=t}^{T} \lambda_k D_{\boldsymbol{\theta}}(\hat{\boldsymbol{x}}_k, k), \tag{8}$$

where $\hat{\boldsymbol{x}}_k$ represents the estimated sample, and $\lambda_k \in \mathbb{R}$ is a weight parameter at timestep $k$. We anticipate that this new denoising prediction can approximate the denoising target and lead to improved sample quality. Some ODE solvers adopt similar formulations to Equation 8 with numerically determined $\{\lambda_k\}_{k=t}^{T}$ (Liu et al., 2021; Lu et al., 2022; Zhang & Chen, 2022).

One challenge with Equation (8) is that the value of the denoising prediction $\tilde{D}_t$ can be unstable and volatile depending on the weights $\{\lambda_k\}_{k=t}^{T}$. This instability is less likely to occur with carefully computed weights in ODE solvers, but convergence is not guaranteed when the weights are optimized through distillation. To generate high-quality samples, the sampling process must follow the true ODE trajectory on which the diffusion models were trained (Liu et al., 2021; Song et al., 2023). In other words, the denoising network might not produce meaningful predictions for samples outside the target manifold of data. This issue has been investigated from various perspectives in recent papers (Xiao et al., 2021; Ning et al., 2023; Li et al., 2023).

In order to avoid these problems, we need to constrain Equation (8) so that it adheres to the previous ODE trajectory. The new prediction $\tilde{D}_t$ can be defined as Equation (9):

$$\tilde{D}_t = D_{\boldsymbol{\theta}}(\hat{\boldsymbol{x}}_t, t) + \sum_{k=t+1}^{T} \lambda_k (D_{\boldsymbol{\theta}}(\hat{\boldsymbol{x}}_t, t) - D_{\boldsymbol{\theta}}(\hat{\boldsymbol{x}}_k, k)) \tag{9}$$

$$\approx D_{\boldsymbol{\theta}}(\hat{\boldsymbol{x}}_t, t) + \lambda_t (D_{\boldsymbol{\theta}}(\hat{\boldsymbol{x}}_t, t) - D_{\boldsymbol{\theta}}(\hat{\boldsymbol{x}}_{t+1}, t+1)). \tag{10}$$

Furthermore, we empirically find that using only the denoising output from the previous timestep is sufficient for distilling knowledge from the teacher sampling. Hence, we apply a first-order approximation to obtain Equation (10). The mean of the new denoising prediction approximates that of the original denoising output since the mean does not change significantly between timesteps $t$ and $t+1$ (e.g., $\mathbb{E}[\tilde{D}_t] \approx \mathbb{E}[D_{\boldsymbol{\theta}}(\hat{\boldsymbol{x}}_t, t)]$). This is a key feature of D-ODE solvers, as we aim to remain on the same sampling trajectory as ODE solvers. In Section 5, we visually demonstrate that D-ODE solvers can globally follow the trajectory of existing ODE solvers.

In conclusion, $\tilde{D}_t$ can replace existing outputs of the denoising network to build D-ODE solvers. If ODE solvers already have their own rules for obtaining new denoising predictions, we can calculate Equation (10) with their new predictions instead of the denoising output (e.g., $D_{\boldsymbol{\theta}}(\hat{\boldsymbol{x}}_t, t)$). Specific applications of D-ODE solvers can be found in Appendix D.1 and D.2. Additionally, we also compare different formulations of D-ODE solvers in Appendix D.4.

## 3.3 DISTILLATION OF D-ODE SOLVER

Each denoising step in diffusion models typically comprises two parts: (1) a denoising network $D_{\boldsymbol{\theta}}$ and (2) an ODE solver $S$. Given an estimated noisy sample $\hat{\boldsymbol{x}}_t$ at timestep $t$, the denoising network produces a denoising output $\hat{\boldsymbol{d}}_t = D_{\boldsymbol{\theta}}(\hat{\boldsymbol{x}}_t, t)$, and the ODE solver subsequently generates the next

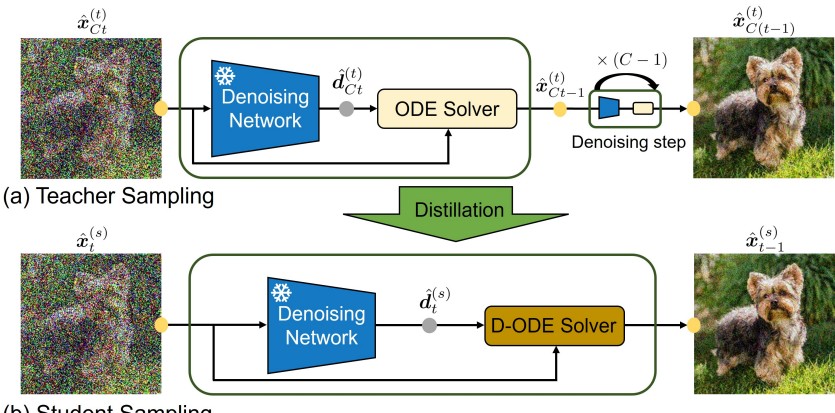

Figure 2: The distillation of ODE Solver. $C$ steps of teacher sampling are distilled into a single step of student sampling. D-ODE Solver $S_d$ is equipped with parameter $\lambda_t$ to be optimized through distillation while denoising network $D_{\boldsymbol{\theta}}$ remains frozen for both teacher and student sampling.

sample $\hat{\boldsymbol{x}}_{t-1} = S(\hat{\boldsymbol{d}}_t, \hat{\boldsymbol{x}}_t)$, utilizing the denoising output and the current noisy sample. While some ODE solvers also utilize the history of denoising outputs $\{\hat{\boldsymbol{d}}_k\}_{k=t}^T$, we omit this notation here for simplicity. This procedure is iterated until the diffusion models reach the estimated original sample $\hat{\boldsymbol{x}}_0$.

Now, we are ready to explain how ODE solvers with large steps can be distilled into D-ODE solvers with small steps. In Figure 2, the teacher sampling process begins with the noisy sample at timestep $Ct$ and undergoes $C$ denoising steps to generate a sample at timestep $C(t-1)$. The student sampling process starts with a noisy sample at timestep $t$ and obtains a sample at timestep $t-1$ after one denoising step. In order to optimize $\lambda_t$ in the D-ODE solver, the teacher sampling is initially performed for one batch to save intermediate samples $\{\hat{\boldsymbol{x}}_k^{(t)}\}_{k=C(t-1)}^{Ct}$ as targets and the student sampling is also executed to acquire intermediate samples $\{\hat{\boldsymbol{x}}_k^{(s)}\}_{k=t-1}^t$ as predictions. Then, $\lambda_t^*$ is determined by minimizing the difference between the targets and the predictions on batch $B$ as follows:

$$\lambda_t^* = \arg\min_{\lambda_t} \mathbb{E}_{\boldsymbol{x} \in B} ||\hat{\boldsymbol{x}}_{C(t-1)}^{(t)} - S_d(D_{\boldsymbol{\theta}}(\hat{\boldsymbol{x}}_t^{(s)}, t), \hat{\boldsymbol{x}}_t^{(s)}; \lambda_t)||^2 \tag{11}$$

$$= \arg\min_{\lambda_t} \mathbb{E}_{\boldsymbol{x} \in B} ||\hat{\boldsymbol{x}}_{C(t-1)}^{(t)} - \hat{\boldsymbol{x}}_{t-1}^{(s)}||^2. \tag{12}$$

The above equation is solved for every timestep $t$ of the student sampling, yielding a set of optimal $\lambda_t$ values (e.g., $\lambda^* = \{\lambda_1^*, \lambda_2^*, ..., \lambda_T^*\}$). Notably, $\lambda^*$ is estimated using only one batch of samples, a process that typically takes just a few CPU minutes, and can be reused for other batches later.

Algorithm 1 outlines the overall sampling procedure of the D-ODE solver. When aiming to generate $N$ samples, it is customary to divide $N$ into $M$ batches and sequentially execute the sampling process for each batch $B$, which contains $|B| = N/M$ samples (Line 3). For the first batch, the teacher sampling is performed with denoising network $D_{\boldsymbol{\theta}}$ and ODE solver $S$ for $CT$ steps to obtain intermediate outputs, which will serve as target samples (Line 5). Subsequently, the student sampling takes place for $T$ steps with $\lambda$ (Line 6). At this point, $\lambda^*$ is estimated and saved for each timestep by solving Equation (12) (Line 7). Starting from the second batch onwards, sampling can proceed using the same denoising network $D_{\boldsymbol{\theta}}$ and D-ODE solver $S_d$ equipped with $\lambda^*$ (Line 9). It is important to note that the student's samples can be generated in just $T$ steps, which should exhibit similar quality to the teacher's samples generated over $CT$ steps. The scale $C$ and batch size $|B|$ are integer values determined prior to experiments, with ablation studies on these parameters presented in Appendix F.

---

**Algorithm 1** Sampling with D-ODE solver

---

1: Pre-trained denoising network $D_{\boldsymbol{\theta}}$, ODE solver $S$, D-ODE solver $S_d$
2: Number of batches $M$ with size $|B|$, Student sampling steps $T$, Teacher sampling steps $CT$
3: **for** $m = 1, ..., M$ **do**
4:     **if** $m = 1$ **then**
5:         $\{\hat{\boldsymbol{x}}_k^{(t)}\}_{k=0}^{CT} = Sampling(D_\theta, S, CT)$          $\triangleright$ Obtain teacher samples
6:         $\{\hat{\boldsymbol{x}}_k^{(s)}\}_{k=0}^{T} = Sampling(D_\theta, S_d, T; \lambda)$         $\triangleright$ Obtain student samples
7:         Estimate $\lambda^* = \{\lambda_1^*, \lambda_2^*, ..., \lambda_T^*\}$ with Equation (12)
8:     **end if**
9:     $\{\hat{\boldsymbol{x}}_k^{(s)}\}_{k=0}^{T} = Sampling(D_\theta, S_d, T; \lambda^*)$
10:    Save sample $\hat{\boldsymbol{x}}_0^{(s)}$
11: **end for**

---

## 4 EXPERIMENTS

In this section, we compare D-ODE solvers to ODE solvers across diverse image generation benchmarks at various resolutions, including CIFAR-10 ($32 \times 32$), CelebA ($64 \times 64$), ImageNet ($64 \times 64$ and $128 \times 128$), FFHQ ($64 \times 64$), and LSUN bedroom ($256 \times 256$). We carry out comprehensive experiments for both noise and data prediction models, each involving a distinct set of ODE solvers. The Fréchet Inception Distance (FID) (Heusel et al., 2017) is measured with 50K generated samples across various numbers of denoising function evaluations (NFE), following Lu et al. (2022). Our reported FID scores are averages from three independent experiment runs with different random seeds.

For the distillation of ODE solvers, we opt for a scale parameter of $C = 10$ and a batch size of $|B| = 100$. However, due to GPU memory constraints in the case of LSUN bedroom, we use a batch size of 25. It is noteworthy that, unless explicitly specified, DDIM serves as the primary teacher sampling method for guiding the student sampling. This choice is informed by the consideration that certain ODE solvers employ multi-step approaches during sampling, making it challenging to set their intermediate outputs as targets for distillation. In contrast, DDIM generates a single intermediate output per denoising step, simplifying the establishment of matching pairs between DDIM targets and student predictions. More experimental details can be found in Appendix E.

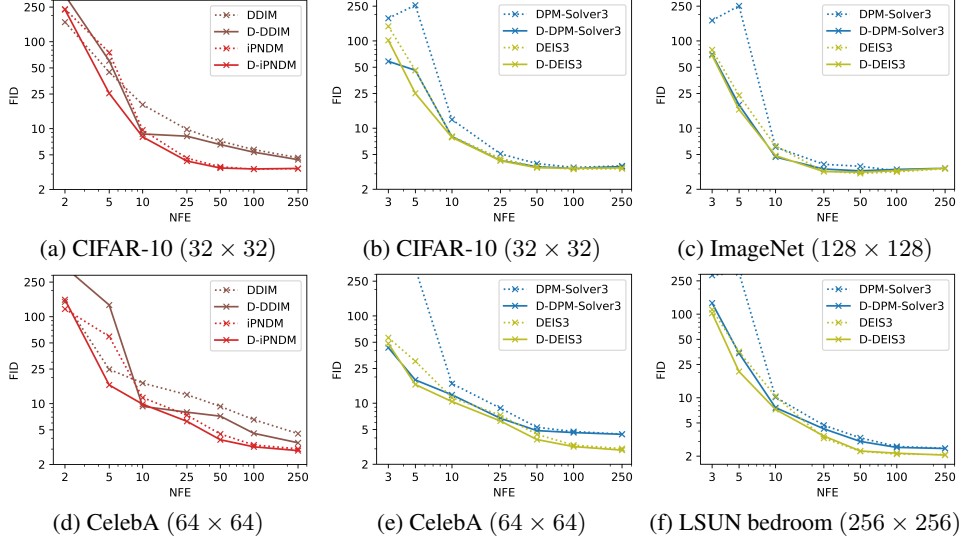

Figure 3: Image quality measured by FID $\downarrow$ with NFE $\in \{2, 5, 10, 25, 50, 100, 250\}$. For DPM-Solver3 and DEIS3, we use 3 NFE instead of 2 NFE as the third-order method requires at least three denoising outputs. Dotted lines denote ODE solvers while straight lines represent the applications of the D-ODE solver to them. More experiments can be found in Appendix H.

## 4.1 Noise prediction model

We apply D-ODE solvers to discrete-time ODE solvers employed in the noise prediction model, which includes DDIM (Song et al., 2020a), iPNDM (Zhang & Chen, 2022), DPM-Solver (Lu et al., 2022), and DEIS (Zhang & Chen, 2022). For DPM-Solver and DEIS, we selected third-order methods. While these prior ODE solvers were primarily evaluated with NFE greater than 10, we also conducted experiments with extremely small NFE such as 2 or 3, to assess the performance of D-ODE solvers during the initial stages of the sampling process.

As shown in Figure 3a and 3d, D-DDIM outperforms DDIM when NFE exceeds 5, gradually converging to FID score similar to that of DDIM as NFE increases. It is important to note that DDIM with small NFE (2 or 5) lacks the capability to produce meaningful images, which is also reflected in the performance of D-DDIM. iPNDM, a high-order method that utilizes previous denoising outputs, consistently exhibits improvements with the D-ODE solver formulation, except at 2 NFE. This improvement is particularly notable for high-order methods like DPM-Solver3 and DEIS3. Specifically, D-DPM-Solver3 effectively alleviates the instability associated with multi-step approaches at extremely small NFE values, surpassing the performance of DPM-Solver3 by a significant margin. While DEIS3 already provides a precise representation of the current denoising prediction through high-order approximation, Figure 3 illustrates that D-DEIS3 can further enhance the approximation through distillation.

## 4.2 Data prediction model

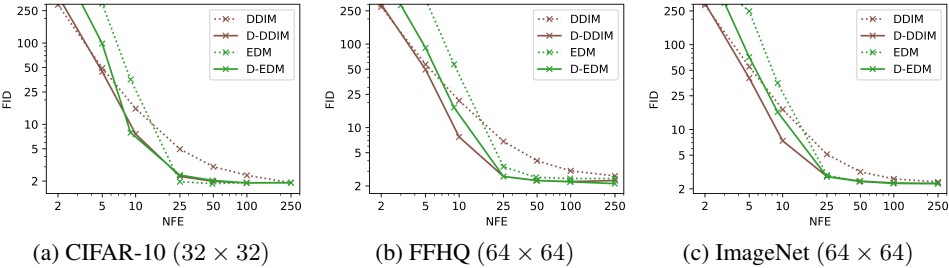

(a) CIFAR-10 (32 × 32)    (b) FFHQ (64 × 64)    (c) ImageNet (64 × 64)

Figure 4: Image quality measured by FID ↓ with various NFE values (DDIM: {2, 5, 10, 25, 50, 100, 250} and EDM: {3, 5, 9, 25, 49, 99, 249}). Dotted lines denote ODE solvers and straight lines represent the applications of the D-ODE solver to them. D-ODE solvers outperform ODE solvers, especially for smaller NFE.

To conduct experiments on data prediction models, we followed the configuration outlined by Karras et al. (2022). We applied the D-ODE solver to DDIM, rebuilt based on this configuration, and EDM (Karras et al., 2022), which employs Heun's second-order method. Notably, while Karras et al. (2022) also re-implemented Euler-based samplers in their paper, we chose not to include them in our experiments, as EDM demonstrates superior FID scores.

As depicted in Figure 4, D-ODE solvers outperform ODE solvers, especially for smaller NFE. For instance, D-DDIM with 25 NFE can produce samples comparable to DDIM with 250 NFE in terms of FID, resulting in a speedup of around 10 times. With increasing NFE, FID scores of both ODE and D-ODE solvers asymptotically converge to each other. Given that the performance of student sampling is closely tied to that of teacher sampling, it is natural to observe similar FID scores for student and teacher sampling with larger NFE. Moreover, it is worth noting that around NFE 2, DDIM occasionally outperforms D-DDIM slightly. This observation suggests that the 2-step DDIM may not possess sufficient capacity to effectively distill knowledge from teacher sampling, particularly when DDIM is already generating noisy images (FID score exceeding 250).

## 4.3 comparison with previous distillation methods

The distillation process for D-ODE solvers typically requires only a few CPU minutes, adding negligible computational overhead to the entire sampling process. In contrast, previous distillation techniques for diffusion models (Salimans & Ho, 2021; Meng et al., 2023; Song et al., 2023) neces-

sitate the optimization of the entire parameters of the denoising network. As a result, these methods demand a substantial amount of training time for each setting.

Table 1 directly compares the computational times required by each distillation method to reach 3 FID on CIFAR-10 given the same pre-trained denoising network. The total time encompasses the distillation time following their configurations and the time taken for generating 50k samples. For instance, D-EDM first optimizes $\lambda$ and then proceeds with the sampling process, while consistency distillation (CD) (Song et al., 2023) and progressive distillation (PD) (Salimans & Ho, 2021) need numerous training iterations before executing a few-step sampling.

The results clearly demonstrate that optimizing ODE solvers instead of the denoising network can significantly reduce computational time and resource requirements while achieving comparable sample quality. It is important to note that the results may vary depending on the training configuration of CD and PD, as the majority of their time is consumed during the distillation process. In this context, our method aligns well with the recent trend

Table 1: Comparison on computational time to achieve 3 FID. The unit of time corresponds to the time required to generate 50k samples with 10-step DDIM.

| Method | D-EDM | CD | PD |
|--------|-------|--------|--------|
| Time | 2.55 | 187.25 | 106.16 |

of democratizing diffusion models by minimizing or circumventing extensive training that relies on a large number of GPUs (Hang et al., 2023; Wang et al., 2023; Zheng et al., 2023; Wu et al., 2023). Appendix C provides a detailed explanation for distillation methods in diffusion models and Appendix G contains additional comparisons with other sampling methods.

## 5 ANALYSIS

This section includes visualizations of the sampling process and qualitative results. We begin by conducting a visual analysis following the methodology of Liu et al. (2021) to assess the global and local characteristics of the sampling process. Subsequently, we compare the quality of the generated images produced by ODE solvers and D-ODE solvers.

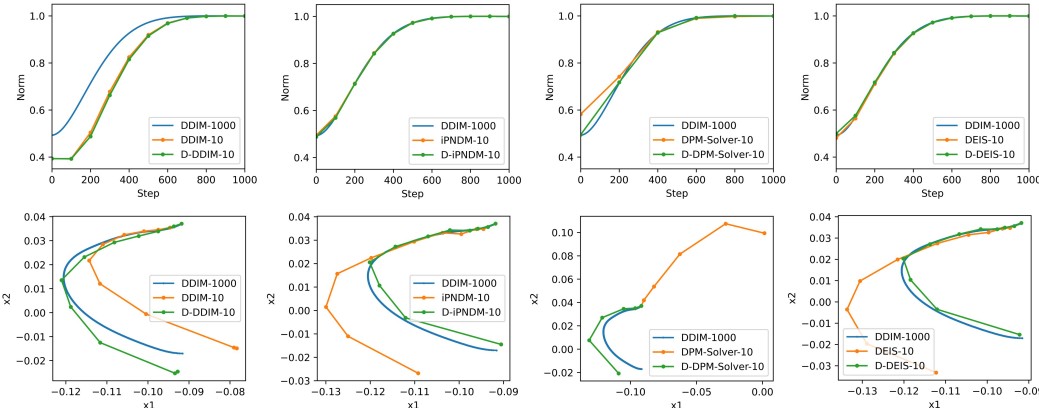

Figure 5: The top row illustrates the change of norm comparing ODE and D-ODE solvers. The bottom row presents the update path of two randomly selected pixels in the images. The result of 1000-step DDIM is drawn as the target trajectory and 10-step sampler is conducted for ODE and D-ODE solvers. The figures are generated from 1000 samples using a noise prediction model trained on CIFAR-10.

To facilitate the visualization of high-dimensional data, we employ two distinct measures: the change in norm as a global feature and the change in specific pixel values as a local feature, as proposed by Liu et al. (2021). In the top row of Figure 5, we observe that the norm of D-ODE solvers closely follows the trajectory traced by the norm of ODE solvers. This observation suggests that D-ODE solvers remain within the high-density regions of the data, exerting minimal influence on the ODE trajectory. This aligns with our design objective of D-ODE solvers, ensuring that the new denoising prediction should match the mean of the denoising output of ODE solvers, as discussed in Section 3.2. For reference, we also include the norm of DDIM with 1000 steps as it

adheres to the target data manifold. In the bottom row of Figure 5, we randomly select two pixels from the image and depict the change in their values, referencing the 1000-step DDIM as the target. Clearly, the pixel values of D-ODE solvers exhibit closer proximity to the target than those of ODE solvers. In conclusion, D-ODE solvers can achieve high-quality image generation by guiding their pixels toward the desired targets while remaining faithful to the original data manifold.

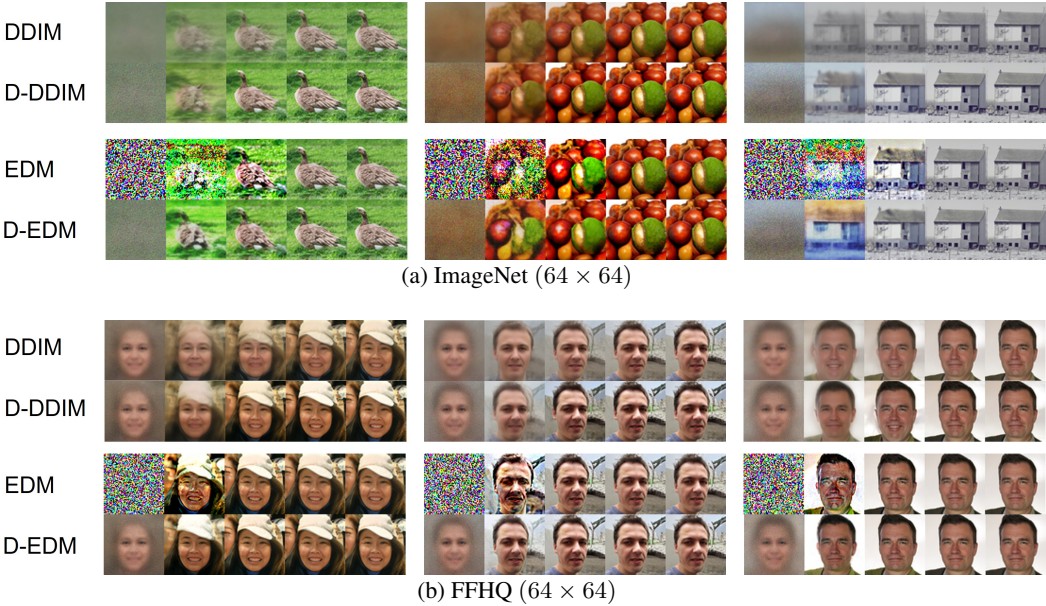

(a) ImageNet ($64 \times 64$)

(b) FFHQ ($64 \times 64$)

Figure 6: Comparison of generated samples between ODE and D-ODE solvers. Data prediction models are used with increasing NFE (DDIM: $\{2, 5, 10, 25, 50\}$, EDM: $\{3, 5, 9, 25, 49\}$).

In Figure 6, we present a comparison of the generated images produced by ODE and D-ODE solvers using data prediction models trained on the ImageNet and FFHQ datasets. In general, our method exhibits an improvement in image quality over ODE solvers, particularly for smaller NFE. DDIM tends to generate blurry images with indistinct boundaries, while D-DDIM produces clearer images with more prominent color contrast. EDM, especially with NFE smaller than 5, generates images characterized by high noise levels and artifacts, leading to FID scores exceeding 250. In contrast, D-EDM manages to generate relatively clear objects even at 5 NFE. More qualitative results can be found in Appendix H.

## 6 Conclusion and Limitations

In this work, we have introduced D-ODE solvers, a novel distillation method tailored for diffusion models. D-ODE solvers are simply formulated by adding a single parameter to ODE solvers. They efficiently distill knowledge from teacher sampling with larger steps into student sampling with smaller steps, leveraging the sampling dynamics of existing ODE solvers. Our experiments have demonstrated the effectiveness of D-ODE solvers in improving FID scores of state-of-the-art ODE solvers, particularly for scenarios involving smaller NFE. Furthermore, through visual analysis, we have gained insights into both the global and local features of our method and have observed significant improvements in image quality.

However, the magnitude of improvement tends to be marginal or limited for large NFE values, ultimately converging to the FID score of the teacher sampling process. Nevertheless, D-ODE solvers remain an attractive option for enhancing sample quality with negligible additional computational cost. Moreover, for the generation of high-resolution images, D-ODE solvers may not be sufficient on their own, as they are parameterized by a single parameter. To better capture the intricate relationships among denoising outputs, one may explore the use of local-specific parameters, achieved by dividing images into smaller grids or by working within the latent space of samples (Rombach et al., 2022). While these possibilities are intriguing, we leave them for future work.

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

## A    TRILEMMA OF GENERATIVE MODELS

Generative models face a trilemma characterized by three essential components, as outlined by Xiao et al. (2021):

1. **High-Quality Samples**: Generative models should demonstrate the capacity to produce high-quality samples.

2. **Mode Coverage and Sample Diversity**: They ought to exhibit mode coverage, ensuring that generated samples are diverse and encompass various modes within the data distribution.

3. **Fast Sampling**: Efficient generative models should possess the ability to generate samples rapidly.

For instance, generative adversarial networks (GANs) (Goodfellow et al., 2014; Brock et al., 2018) excel in generating high-quality samples with just a single evaluation of the network. Nevertheless, GANs often struggle with generating diverse samples, resulting in poor mode coverage (Salimans et al., 2016; Zhao et al., 2018). Conversely, Variational Autoencoders (VAEs) (Kingma & Welling, 2013) and Normalizing Flows (Dinh et al., 2016) are designed to adequately ensure mode coverage but may suffer from low sample quality. Recently, diffusion models have emerged as a novel class of generative models that can generate high-quality samples comparable to GANs (Dhariwal & Nichol, 2021; Saharia et al., 2022), while also providing a rich variety of samples. However, conventional diffusion models often require hundreds to thousands of network evaluations for sampling, rendering them computationally expensive in practice. The primary bottleneck in the sampling process of diffusion models is closely tied to the number of denoising network evaluations. Consequently, numerous research works have explored techniques to expedite the sampling process by either skipping or optimizing sampling steps while maintaining the quality of generated samples. These techniques can be broadly classified into two categories: learning-based and learning-free sampling methods (Yang et al., 2022a) as introduced in Section 1.

## B    NOISE AND DATA PREDICTION MODELS

The output of the denoising network should be parameterized to estimate the score function referring to the reverse-time ODE in Equation 5. The score function represents the gradient of the logarithm of the data distribution, indicating the direction of data with higher likelihood and less noise. One straightforward approach for the parameterization is to directly estimate the original data $\boldsymbol{x}$, in which case the score function is estimated by calculating the gradient toward the original data given the current noise level:

$$\nabla_{\boldsymbol{x}} \log q_t(\boldsymbol{x}_t) = \frac{\hat{x}_{\boldsymbol{\theta}}(\boldsymbol{x}_t, t) - \boldsymbol{x}_t}{\sigma_t^2}. \tag{13}$$

Another approach indirectly designs the denoising network to predict noise $\epsilon$, which represents the residual signal infused in the original sample. In this case, the score function can be calculated as:

$$\nabla_{\boldsymbol{x}} \log q_t(\boldsymbol{x}_t) = -\frac{\hat{\epsilon}_{\boldsymbol{\theta}}(\boldsymbol{x}_t, t)}{\sigma_t}. \tag{14}$$

While both noise and data prediction models are theoretically equivalent (Kingma et al., 2021; Luo, 2022; Karras et al., 2022), they reveal different characteristics during the sampling process.

**Noise prediction models**  Noise prediction models may initially introduce significant discrepancies between the ground truth noise and the predicted noise Benny & Wolf (2022). Since sampling commences with highly noisy samples, the denoising network lacks sufficient information to accurately predict noise Ho et al. (2020). Additionally, the magnitude of correction required at each timestep is relatively small, necessitating multiple timesteps to rectify such deviations (Luo, 2022).

**Data prediction models**  Data prediction models are known to offer better accuracy in the initial stages of sampling, while noise prediction models become preferable in later stages. Predicting data assists the denoising network in understanding the global structure of the target sample (Luo, 2022). Empirical evidence shows that the predicted data is close to the ground truth at the beginning of the sampling procedure (Ramesh et al., 2022; Guan et al., 2022). However, in the later stages when substantial structures have already been formed and only minor noise artifacts need to be removed, finer details become challenging to recover (Benny & Wolf, 2022). Essentially, the information provided by early data prediction becomes less effective in the later stages of sampling.

**Our experiments**  The difference between data and noise prediction models is also evident in our experiments, as depicted in Figure 1. Predictions of $\epsilon$ in the initial sampling stages exhibit higher correlation with each other than those in later stages, whereas predictions of $\boldsymbol{x}$ become more correlated in the later stages compared to the earlier stages. In the case of noise estimation, a small amount of noise remains in a sample for the last few timesteps, resulting in detailed and minor changes to the sample with high variance. In conclusion, different details are modified at each timestep during the later sampling process.

On the other hand, it is challenging for a $\boldsymbol{x}$ estimator to predict the original sample from the initial noisy sample. However, its predictions become more consistent in the later stages of sampling as the sample becomes less noisy. This observation aligns with the analysis presented in Benny & Wolf (2022), which indicates that the variance of the $\boldsymbol{x}$ estimator gradually decreases with more sampling steps, while the variance of the $\epsilon$ estimator abruptly increases in the last phase of sampling.

## C  KNOWLEDGE DISTILLATION IN DIFFUSION MODELS

Knowledge distillation (Hinton et al., 2015) was initially introduced to transfer knowledge from a larger model (teacher) to a smaller one (student), with the student model being trained to imitate the output of the teacher model. This concept can be applied to diffusion-based sampling processes to merge several timesteps (teacher) into a single timestep (student) to accelerate generation speed.

Luhman & Luhman (2021) directly apply knowledge distillation to diffusion models by minimizing the difference between the outputs of a one-step student sampler and the outputs of a multi-step DDIM sampler. Thus, the student model is trained to imitate the output of the teacher model, being initialized with a pre-trained denoising network to inherit knowledge from the teacher.

Subsequently, progressive distillation (Salimans & Ho, 2021) proposes an iterative approach to train a student network to merge two sampling timesteps of the teacher network until it achieves one-step sampling to imitate the entire sampling process. This allows the student network to gradually learn the teacher's sampling process, as learning to predict the output of two-step sampling is easier than learning to predict the output of multi-step sampling. Given a pre-trained denoising network $\boldsymbol{\theta}$ as the teacher, Salimans & Ho (2021) first train a student network $\boldsymbol{\theta}'$ to predict the output of two sampling timesteps of the teacher network. The student $\boldsymbol{\theta}'$ then becomes the new teacher and a new student with parameter $\boldsymbol{\theta}''$ is trained to combine two sampling timesteps of the new teacher network $\boldsymbol{\theta}'$ until

the total timestep reaches one step. The student model is parameterized and initialized with the same deep neural network as the teacher model, and progressive distillation is examined with the DDIM sampler.

Meng et al. (2023) extend progressive distillation to scenarios involving classifier-free diffusion guidance, achieving single-step or few-step generation for text-to-image generation, class-conditioned generation, image-to-image translation, and image inpainting. They leverage a two-stage approach to train a student model to match the combined output of the conditional and unconditional models first, and then apply progressive distillation by setting the student model as the new teacher. Most of the configuration remains the same as Salimans & Ho (2021), mainly utilizing DDIM sampler.

Recently, Song et al. (2023) proposes a new class of generative models called consistency models which exploit the consistency property on the trajectory of a probabilistic flow ODE. They are trained to predict the original sample from any point on the same ODE trajectory. During training, a target network and an online network are utilized so that the online network is optimized to generate the same output as the target network, while the target network is updated with an exponential moving average. Consistency models can generate samples in a single step or a few steps by design and are also capable of image inpainting, colorization, and super-resolution in a zero-shot fashion. They can be trained either independently or via distillation, which is named as consistency training and consistency distillation, respectively. In this paper, we are interested in consistency distillation in comparison with our distillation method.

However, these distillation methods typically require extensive training to adapt to different pre-trained models, datasets, and ODE solvers, which limits their practical applicability. In this paper, we propose to optimize newly parameterized ODE solvers (D-ODE solvers) exclusively. This approach effectively distills the sampling process with larger steps into a new process with smaller steps while keeping the pre-trained denoising network fixed. Because our method does not require parameter updates for the denoising network, the distillation process can be completed in just a few CPU minutes.

## D  IMPLEMENTATION DETAILS OF D-ODE SOLVERS

In this section, we explain the ODE solvers of our interest in detail and their application in the framework of D-ODE solvers. We categorize ODE solvers into two distinct types based on the nature of the diffusion timestep: discrete and continuous. Discrete-time ODE solvers include DDIM, PNDM, DPM-Solver, and DEIS, where we built our code upon Lu et al. (2022), while continuous-time ODE solvers contain re-implementations of DDIM and EDM based on the work done by Karras et al. (2022).

### D.1  D-ODE SOLVERS IN NOISE PREDICTION MODELS

DDIM (Song et al., 2020a) is formulated as a non-Markovian diffusion process of DDPM (Ho et al., 2020), defining a deterministic generation procedure using implicit models. Given the estimated sample $\hat{\boldsymbol{x}}_t$ at timestep $t$, their sampling process is expressed as follows:

$$\hat{\boldsymbol{x}}_{t-1} = \alpha_{t-1} \underbrace{\left( \frac{\hat{\boldsymbol{x}}_t - \sigma_t \epsilon_{\boldsymbol{\theta}}(\hat{\boldsymbol{x}}_t, t)}{\alpha_t} \right)}_{\text{predicted } x_0} + \sigma_{t-1} \underbrace{\epsilon_{\boldsymbol{\theta}}(\hat{\boldsymbol{x}}_t, t)}_{\text{direction toward } \boldsymbol{x}_t}. \tag{15}$$

Here, $(\alpha_t, \sigma_t)$ represents a predefined noise schedule. With the noise prediction model $\epsilon_{\boldsymbol{\theta}}$, the new denoising prediction $\tilde{D}_t$, formulated by D-ODE solver, is defined as $\tilde{D}_t = \epsilon_{\boldsymbol{\theta}}(\hat{\boldsymbol{x}}_t, t) + \lambda_t(\epsilon_{\boldsymbol{\theta}}(\hat{\boldsymbol{x}}_t, t) - \epsilon_{\boldsymbol{\theta}}(\hat{\boldsymbol{x}}_{t+1}, t+1))$, referring to Equation (10). We then simply substitute the denoising prediction into the sampling equation:

$$\hat{\boldsymbol{x}}_{t-1} = \alpha_{t-1} \left( \frac{\hat{\boldsymbol{x}}_t - \sigma_t \tilde{D}_t}{\alpha_t} \right) + \sigma_{t-1} \tilde{D}_t. \tag{16}$$

This equation defines D-DDIM with $\lambda_t$ to be optimized through distillation. In cases where the previous denoising output is unavailable (e.g., at timestep $T$), we use the given noisy sample to define the denoising prediction. This results in $\tilde{D}_T^{(0)} = \epsilon_{\boldsymbol{\theta}}(\hat{\boldsymbol{x}}_T, T) + \lambda_T(\epsilon_{\boldsymbol{\theta}}(\hat{\boldsymbol{x}}_T, T) - \boldsymbol{x}_T)$ at initial timestep $T$. The assumption that both $\boldsymbol{x}_T$ and $\epsilon_{\boldsymbol{\theta}}(\hat{\boldsymbol{x}}_T, T)$ follow a Normal distribution $\mathcal{N}(\boldsymbol{0}, \sigma_t^2 \boldsymbol{I})$ in theory ensures that the mean of the denoising prediction remains consistent with the original denoising output. It is expected that $(\epsilon_{\boldsymbol{\theta}}(\hat{\boldsymbol{x}}_T, T) - \boldsymbol{x}_T)$ contains information regarding the direction toward the true $\boldsymbol{x}_{T-1}$ to some extent, which actually improves the FID score in practice. Thus, we also apply this sampling recipe to other D-ODE solvers based on noise prediction models.

PNDM (Liu et al., 2021) is based on pseudo-numerical methods on the data manifold, built upon the observation that classical numerical methods can deviate from the high-density area of data. PNDM encapsulates DDIM as a simple case and surpasses DDIM with its high-order methods. However, PNDM requires 12 NFE for the first 3 steps, making it challenging to compare with other methods using a fixed NFE. Therefore, we opt for iPNDM (Zhang & Chen, 2022), which eliminates the need for initial warm-up steps and outperforms PNDM while maintaining the pseudo-numerical sampling process. iPNDM employs a linear combination of multiple denoising outputs to represent the current denoising output while adhering to the sampling update path of DDIM, as shown below:

$$\hat{\epsilon}_t^{(3)} = \frac{1}{24}(55\epsilon_{\boldsymbol{\theta}}(\hat{\boldsymbol{x}}_t, t) - 59\epsilon_{\boldsymbol{\theta}}(\hat{\boldsymbol{x}}_{t+1}, t+1) + 37\epsilon_{\boldsymbol{\theta}}(\hat{\boldsymbol{x}}_{t+2}, t+2) - 9\epsilon_{\boldsymbol{\theta}}(\hat{\boldsymbol{x}}_{t+3}, t+3)), \quad (17)$$

$$\hat{\boldsymbol{x}}_{t-1} = \alpha_{t-1}\left(\frac{\hat{\boldsymbol{x}}_t - \sigma_t\hat{\epsilon}_t^{(3)}}{\alpha_t}\right) + \sigma_{t-1}\hat{\epsilon}_t^{(3)}, \quad (18)$$

where $\hat{\epsilon}_t$ is approximated with three previous denoising outputs and then applied to DDIM sampling. Therefore, the first three denoising outputs should be defined independently as follows:

$$\hat{\epsilon}_t^{(0)} = \epsilon_{\boldsymbol{\theta}}(\hat{\boldsymbol{x}}_t, t), \quad (19)$$

$$\hat{\epsilon}_t^{(1)} = \frac{3}{2}\epsilon_{\boldsymbol{\theta}}(\hat{\boldsymbol{x}}_t, t) - \frac{1}{2}\epsilon_{\boldsymbol{\theta}}(\hat{\boldsymbol{x}}_{t+1}, t+1), \quad (20)$$

$$\hat{\epsilon}_t^{(2)} = \frac{1}{12}(23\epsilon_{\boldsymbol{\theta}}(\hat{\boldsymbol{x}}_t, t) - 16\epsilon_{\boldsymbol{\theta}}(\hat{\boldsymbol{x}}_{t+1}, t+1) + 5\epsilon_{\boldsymbol{\theta}}(\hat{\boldsymbol{x}}_{t+2}, t+2)). \quad (21)$$

Leveraging these newly defined denoising outputs $\hat{\epsilon}_t$, we construct D-iPNDM, where the new denoising prediction $\tilde{D}_t$ can be defined as $\tilde{D}_t = \hat{\epsilon}_t + \lambda_t(\hat{\epsilon}_t - \hat{\epsilon}_{t+1})$. This leads to the same update rule in Equation (16).

DPM-Solver (Lu et al., 2022) leverages the semi-linear structure of probabilistic flow ODEs by solving the exact formulation of the linear part of ODEs and approximating the weighted integral of the neural network with exponential integrators (Hochbruck & Ostermann, 2010). DPM-Solver offers first-order, second-order, and third-order methods, with the first-order variant corresponding to DDIM. DPM-Solver strategically divides the total sampling steps using these different-order methods. For instance, DPM-Solver2 (second-order DPM-Solver) is employed 5 times to generate a sample comprising 10 denoising steps, with the denoising network being evaluated twice within DPM-Solver2. To achieve 15 denoising steps, DPM-Solver2 is applied 7 times, and DPM-Solver1 (or DDIM) is applied during the final denoising step.

In this section, we delve into the formulation of D-DPM-Solver2, and the application to DPM-Solver3 follows a similar approach. First, we denote $\tau_t = \log(\alpha_t/\sigma_t)$ as the logarithm of the signal-to-noise ratio (SNR), and $\tau_t$ is a strictly decreasing function as $t$ increases. Consequently, we can establish an inverse function mapping from $\tau$ to $t$, denoted as $t_\tau(\cdot) : \mathbb{R} \to \mathbb{R}$. Now, we can outline DPM-Solver2 with the following steps:

$$t - \frac{1}{2} = t_\tau(\frac{\tau_{t-1} + \tau_t}{2}), \quad (22)$$

$$\hat{\boldsymbol{x}}_{t-\frac{1}{2}} = \frac{\alpha_{t-\frac{1}{2}}}{\alpha_t}\hat{\boldsymbol{x}}_t - \sigma_{t-\frac{1}{2}}(e^{\frac{h_t}{2}} - 1)\hat{\epsilon}_{\boldsymbol{\theta}}(\hat{\boldsymbol{x}}_t, t), \quad (23)$$

$$\hat{\boldsymbol{x}}_{t-1} = \frac{\alpha_{t-1}}{\alpha_t}\hat{\boldsymbol{x}}_t - \sigma_{t-1}(e^{h_t} - 1)\hat{\epsilon}_{\boldsymbol{\theta}}(\hat{\boldsymbol{x}}_{t-\frac{1}{2}}, t - \frac{1}{2}). \quad (24)$$

In these equations, $h_t = \tau_{t-1} - \tau_t$, and $\hat{\boldsymbol{x}}_{t-\frac{1}{2}}$ represents the intermediate output between timestep $t - 1$ and $t$. Since DPM-Solver2 utilizes a two-stage denoising step, we must define two denoising

predictions to formulate D-DPM-Solver2 with $\lambda_t$ and $\lambda_{t-\frac{1}{2}}$ optimized through distillation:

$$\tilde{D}_t = \hat{\epsilon}_{\boldsymbol{\theta}}(\hat{\boldsymbol{x}}_t, t) + \lambda_t(\hat{\epsilon}_{\boldsymbol{\theta}}(\hat{\boldsymbol{x}}_t, t) - \hat{\epsilon}_{\boldsymbol{\theta}}(\hat{\boldsymbol{x}}_{t+\frac{1}{2}}, t + \frac{1}{2})), \tag{25}$$

$$\tilde{D}_{t-\frac{1}{2}} = \hat{\epsilon}_{\boldsymbol{\theta}}(\hat{\boldsymbol{x}}_{t-\frac{1}{2}}, t - \frac{1}{2}) + \lambda_{t-\frac{1}{2}}(\hat{\epsilon}_{\boldsymbol{\theta}}(\hat{\boldsymbol{x}}_{t-\frac{1}{2}}, t - \frac{1}{2}) - \hat{\epsilon}_{\boldsymbol{\theta}}(\hat{\boldsymbol{x}}_t, t)). \tag{26}$$

These predictions are then used in Equation (23) and (24), respectively:

$$\hat{\boldsymbol{x}}_{t-\frac{1}{2}} = \frac{\alpha_{t-\frac{1}{2}}}{\alpha_t}\hat{\boldsymbol{x}}_t - \sigma_{t-\frac{1}{2}}(e^{\frac{h_t}{2}} - 1)\tilde{D}_t, \tag{27}$$

$$\hat{\boldsymbol{x}}_{t-1} = \frac{\alpha_{t-1}}{\alpha_t}\hat{\boldsymbol{x}}_t - \sigma_{t-1}(e^{h_t} - 1)\tilde{D}_{t-\frac{1}{2}}. \tag{28}$$

Similar to DPM-Solver, DEIS (Zhang & Chen, 2022) employs an exponential integrator to leverage the semi-linear structure of the reverse-time diffusion process. In particular, they propose the use of high-order polynomials to approximate the non-linear term in ODEs as shown below:

$$P_r(t) = \sum_{j=0}^{r} C_{tj}\hat{\epsilon}_{\boldsymbol{\theta}}(\hat{\boldsymbol{x}}_{t+j}, t + j) \tag{29}$$

$$\hat{\boldsymbol{x}}_{t-1} = \frac{\alpha_{t-1}}{\alpha_t}\hat{\boldsymbol{x}}_t + P_r(t). \tag{30}$$

Here, $\{C_{tj}\}_{j=0}^{r}$ is numerically determined through weighted integration to approximate the true ODE trajectory. DEIS offers several variants based on the numerical method used to estimate $C_{tj}$, and for our experiments, we choose $t$AB-DEIS as it exhibits the most promising results among the variants. Additionally, Zhang & Chen (2022) explores DEIS for different values of $r \in \{1, 2, 3\}$ where larger values of $r$ generally lead to improved approximations of the target denoising prediction. It is worth noting that DDIM can be seen as a special case of $t$AB-DEIS with $r = 0$.

With reference to Equation (30), we define a new denoising prediction $\tilde{D}_t$ and D-DEIS as follows:

$$\tilde{D}_t = P_r(t) + \lambda_t(P_r(t) - P_r(t + 1)), \tag{31}$$

$$\hat{\boldsymbol{x}}_{t-1} = \frac{\alpha_{t-1}}{\alpha_t}\hat{\boldsymbol{x}}_t + \tilde{D}_t. \tag{32}$$

## D.2 D-ODE SOLVERS IN DATA PREDICTION MODELS

In our study, we newly implement DDIM (Song et al., 2020a) in a continuous setting using a data prediction model. We follow the configurations outlined by Karras et al. (2022). The sampling process for this modified DDIM is defined as follows:

$$s_t = \frac{x_{\boldsymbol{\theta}}(\hat{\boldsymbol{x}}_t, t) - \hat{\boldsymbol{x}}_t}{\sigma_t}, \tag{33}$$

$$\hat{\boldsymbol{x}}_{t-1} = \hat{\boldsymbol{x}}_t + (\sigma_t - \sigma_{t-1})s_t, \tag{34}$$

where $s_t$ approximates the score function, directing toward the high-density area of the data, and the denoising step is carried out in Equation (34) based on the difference in noise levels measured by $(\sigma_t - \sigma_{t-1})$. Subsequently, the new denoising prediction of D-DDIM is defined as $\tilde{D}_t = x_{\boldsymbol{\theta}}(\hat{\boldsymbol{x}}_t, t) + \lambda_t(x_{\boldsymbol{\theta}}(\hat{\boldsymbol{x}}_t, t) - x_{\boldsymbol{\theta}}(\hat{\boldsymbol{x}}_{t+1}, t + 1))$. This prediction is then incorporated into the sampling equation for DDIM with the data prediction model as follows:

$$\tilde{s}_t = \frac{\tilde{D}_t - \hat{\boldsymbol{x}}_t}{\sigma_t}, \tag{35}$$

$$\hat{\boldsymbol{x}}_{t-1} = \hat{\boldsymbol{x}}_t + (\sigma_t - \sigma_{t-1})\tilde{s}_t. \tag{36}$$

Karras et al. (2022) introduced the EDM sampler based on Heun's second-order method, which achieved a state-of-the-art FID score on CIFAR-10 and ImageNet64. They utilized a novel ODE

formulation, parameter selection, and modified neural architectures. The EDM sampling process is shown as follows:

$$s_t = \frac{x_{\boldsymbol{\theta}}(\hat{\boldsymbol{x}}_t, t) - \hat{\boldsymbol{x}}_t}{\sigma_t}, \quad \hat{\boldsymbol{x}}'_{t-1} = \hat{\boldsymbol{x}}_t + (\sigma_t - \sigma_{t-1})s_t, \tag{37}$$

$$s'_t = \frac{x_{\boldsymbol{\theta}}(\hat{\boldsymbol{x}}'_{t-1}, t-1) - \hat{\boldsymbol{x}}'_{t-1}}{\sigma_{t-1}}, \quad \hat{\boldsymbol{x}}_{t-1} = \hat{\boldsymbol{x}}_t + (\sigma_t - \sigma_{t-1})(\frac{1}{2}s_t + \frac{1}{2}s'_t). \tag{38}$$

The first stage of EDM is equivalent to DDIM, and then the score function is more accurately estimated in the second stage by linearly combining two estimations. Notably, 18 steps of EDM sampling correspond to 35 NFE, as one step of EDM involves two network evaluations, and Equation (38) is not computed at the last step. To construct D-EDM, we define two denoising predictions as $\tilde{D}_t = x_{\boldsymbol{\theta}}(\hat{\boldsymbol{x}}_t, t) + \lambda_t(x_{\boldsymbol{\theta}}(\hat{\boldsymbol{x}}_t, t) - x_{\boldsymbol{\theta}}(\hat{\boldsymbol{x}}'_{t+1}, t+1))$ and $\tilde{D}'_t = x_{\boldsymbol{\theta}}(\hat{\boldsymbol{x}}'_{t-1}, t-1) + \lambda_t(x_{\boldsymbol{\theta}}(\hat{\boldsymbol{x}}'_{t-1}, t-1) - x_{\boldsymbol{\theta}}(\hat{\boldsymbol{x}}_t, t))$. Consequently, the sampling steps for D-EDM are described as follows:

$$\tilde{s}_t = \frac{\tilde{D}_t - \hat{\boldsymbol{x}}_t}{\sigma_t}, \quad \hat{\boldsymbol{x}}'_{t-1} = \hat{\boldsymbol{x}}_t + (\sigma_t - \sigma_{t-1})\tilde{s}_t, \tag{39}$$

$$\tilde{s}'_t = \frac{\tilde{D}'_t - \hat{\boldsymbol{x}}'_{t-1}}{\sigma_{t-1}}, \quad \hat{\boldsymbol{x}}_{t-1} = \hat{\boldsymbol{x}}_t + (\sigma_t - \sigma_{t-1})(\frac{1}{2}\tilde{s}_t + \frac{1}{2}\tilde{s}'_t). \tag{40}$$

### D.3 Various interpretations of D-ODE solvers

New denoising prediction $\tilde{D}_t$ in D-ODE solvers is formulated based on the observation that denoising outputs are highly correlated and that it is essential to retain the same mean as the original outputs. We can rewrite the definition of our denoising prediction as follows:

$$\tilde{D}_t = D_{\boldsymbol{\theta}}(\hat{\boldsymbol{x}}_t, t) + \lambda_t \left( D_{\boldsymbol{\theta}}(\hat{\boldsymbol{x}}_t, t) - D_{\boldsymbol{\theta}}(\hat{\boldsymbol{x}}_{t+1}, t+1) \right). \tag{41}$$

The above formulation can be interpreted to calculate interpolation (or extrapolation) between the current and previous denoising outputs to estimate the target output. Therefore, D-ODE solvers can be seen as the process of dynamically interpolating (or extrapolating) the denoising outputs with $\lambda_t$ optimized through distillation. Similarly, Zhang et al. (2023) proposed the use of extrapolation on the current and previous estimates of the original data $\hat{\boldsymbol{x}}_t$. They argued that extrapolating between two predictions includes useful information toward the target data by refining the true mean estimation. Although accurate extrapolation requires grid search for parameter tuning, they demonstrated improvements in the FID of various ODE solvers.

Another interpretation is based on the work of Permenter & Yuan (2023), who matched the denoising process to gradient descent applied to the Euclidean distance function under specific assumptions. They reinterpreted diffusion models using the definition of projection onto the true data distribution and proposed a new sampler by minimizing the error in predicting $\epsilon$ between adjacent timesteps. Their sampler corresponds to D-DDIM with $\lambda_t = 1$ selected via grid search, and it outperforms DDIM and PNDM.

The last interpretation is that D-ODE solvers accelerate the convergence of sample generation in a way similar to how momentum boosts optimization in SGD (Sutskever et al., 2013). Just as SGD with momentum utilizes the history of previous gradients to speed up parameter updates in a neural network, D-ODE solvers leverage previous denoising outputs to accelerate the convergence of sampling. An interesting future direction could explore whether advanced optimizers used in machine learning models (Kingma, 2014; Duchi et al., 2011; Ruder, 2016) can be effectively applied to diffusion models.

### D.4 Various formulations of D-ODE solvers

To validate D-ODE solvers, we explore different formulations of D-ODE solvers. For example, we can estimate parameters for each denoising output separately instead of optimizing a single parameter $\lambda_t$, which we name D-DDIM-Sep. Additionally, we use the first-order equation in Equation (9)

to achieve our denoising prediction. We also formulate a second-order approximation, which we call D-DDIM2. All methods are presented below for comparison, with $\boldsymbol{d}_t = D_{\boldsymbol{\theta}}(\hat{\boldsymbol{x}}_t, t)$:

$$\text{DDIM}: \quad \boldsymbol{d}_t, \tag{42}$$

$$\text{D-DDIM}: \quad \tilde{D}_t = \boldsymbol{d}_t + \lambda_t(\boldsymbol{d}_t - \boldsymbol{d}_{t+1}), \tag{43}$$

$$\text{D-DDIM-Sep}: \quad \tilde{D}_t = \boldsymbol{d}_t + \lambda_{t1}\boldsymbol{d}_t + \lambda_{t2}\boldsymbol{d}_{t+1}, \tag{44}$$

$$\text{D-DDIM2}: \quad \tilde{D}_t = \boldsymbol{d}_t + \lambda_{t1}(\boldsymbol{d}_t - \boldsymbol{d}_{t+1}) + \lambda_{t2}(\boldsymbol{d}_t - \boldsymbol{d}_{t+2}). \tag{45}$$

Table 2: Comparison on various D-ODE solver formulations. FID is measured on CIFAR-10 with a noise prediction model and the best FID is bolded.

| NFE | 10 | 25 | 50 |
|---|---|---|---|
| DDIM | 18.85 | 9.79 | 7.17 |
| D-DDIM | **8.67** | **8.18** | **6.55** |
| D-DDIM-Sep | 79.21 | 26.40 | 11.50 |
| D-DDIM-2 | 18.75 | 9.83 | 7.21 |

We examined the four formulations mentioned above on CIFAR-10 with different NFE, while all other configurations for distillation and sampling remained the same. The results are summarized in Table 2. Surprisingly, D-DDIM outperforms all other formulations, and D-DDIM-Sep worsens the FID score. This result can be interpreted as the sampling process not converging for D-DDIM-Sep. As we pointed out in Section 3.2, separately estimated parameters may deviate from the target trajectory of ODE solvers. This is due to the fact that $\lambda_{t1}$ and $\lambda_{t2}$, determined by distillation, can be volatile without any constraints and may not reflect the general sampling rules across different batches. D-DDIM2 also does not improve the FID score of DDIM. One possible reason for this is that parameters optimized on one batch may not be applicable to others. Since the two parameters are optimized on only one batch, fine-grained estimation of denoising predictions, like D-DDIM2, may not be valid for all batches.

## E  EXPERIMENT DETAILS

**Model Architectures**  For noise prediction models, we follow the architectures and configurations of Ho et al. (2020); Dhariwal & Nichol (2021), utilizing their pre-trained models. Specifically, we adopt the model architecture and configuration in DDPM (Ho et al., 2020) for experiments on CIFAR-10 and CelebA $64 \times 64$. For ImageNet $128 \times 128$ and LSUN Bedroom $256 \times 256$, we use the corresponding network architecture from Dhariwal & Nichol (2021). In experiments with data prediction models, we utilize the configurations and pre-trained models from Karras et al. (2022) for CIFAR-10, FFHQ $64 \times 64$, and ImageNet $64 \times 64$.

**Distillation Configurations**  As outlined in Algorithm 1, we first perform teacher sampling with $CT$ steps to set target samples, followed by student sampling with $T$ steps to match the student's output with the teacher's targets. For most D-ODE solvers, we use DDIM sampling as the teacher sampling method, as it generates one output per denoising step, enabling one-to-one matching between targets and predictions. For iPNDM and DEIS, we use themselves as the teacher method for distillation, respectively (e.g., DEIS with $CT$ steps as the teacher and D-DEIS with $T$ steps as the student). Although they use a linear combination of previous denoising outputs to estimate current denoising predictions, the sampling dynamics are the same as DDIM. Therefore, the teacher's targets and student's predictions can be easily matched (details in Section D.1).

Moreover, student sampling is performed sequentially to optimize $\lambda$ in D-ODE solvers. In other words, $\lambda_t$ is first estimated via distillation and then the next sample is generated with optimized D-ODE solvers at timestep $t$ during student sampling. This approach helps stabilize the sampling process, as $\lambda_{t+1}$ is estimated based on previously generated samples from D-ODE solvers with $\lambda_t^*$. This can alleviate exposure bias (Ranzato et al., 2016; Ning et al., 2023) with precisely estimated $\lambda$.

**Sampling Details** For simplicity, we adopt uniformly divided timesteps for all ODE solvers. We generate 50K samples and report the mean FID score calculated after three runs with different seeds. All experiments are conducted using GPUs, including NVIDIA TITAN Xp, Nvidia V100, and Nvidia A100. We fix the scale $C = 10$ and batch size $|B| = 100$, except for LSUN Bedroom, where $|B| = 25$ due to memory constraints. Ablation studies on these two parameters are presented in Section F.

Several design choices need to be made for each ODE solver. PNDM requires 12 NFE for the first 3 steps, making it challenging to compare with other methods using a fixed NFE. Therefore, we adopt iPNDM (Zhang & Chen, 2022), which does not require initial warm-up steps and outperforms PNDM. DEIS offers various versions of ODE solvers, among which we select $t$AB-DEIS, exhibiting the best FID score in their experiments. DPM-Solver combines different-order solvers using adaptive step sizes. For simplicity, we opt for the single-step DPM-Solver, which sequentially uses DPM-Solver1, DPM-Solver2, and DPM-Solver3 to compose the total timesteps. While EDM allows stochastic sampling by its design, we employ deterministic sampling to obtain a definite target sample generated by teacher sampling.

# F    ABLATION STUDIES

Table 3: Ablation studies on CIFAR-10 with noise prediction models. Different scale $C \in \{5, 10, 20, 30\}$ and batch size $|B| \in \{5, 10, 50, 100\}$ are examined. We report mean and standard deviation after 3 runs (mean ± std) and the best FID is bolded.

| (a) Different Scale $S$ | | | | (b) Different Batch Size $|B|$ | | | |
|---|---|---|---|---|---|---|---|
| NFE | 10 | 25 | 50 | NFE | 10 | 25 | 50 |
| 5 | $9.68_{\pm 0.10}$ | $8.20_{\pm 0.06}$ | $6.52_{\pm 0.02}$ | 5 | $9.33_{\pm 0.66}$ | $7.75_{\pm 0.13}$ | $6.64_{\pm 0.09}$ |
| 10 | $8.83_{\pm 0.10}$ | $8.09_{\pm 0.03}$ | $6.55_{\pm 0.09}$ | 10 | $8.83_{\pm 0.58}$ | $7.79_{\pm 0.09}$ | $6.55_{\pm 0.07}$ |
| 20 | $8.52_{\pm 0.04}$ | $8.01_{\pm 0.03}$ | $\mathbf{6.50}_{\pm 0.01}$ | 50 | $\mathbf{8.03}_{\pm 0.08}$ | $7.69_{\pm 0.08}$ | $6.58_{\pm 0.05}$ |
| 30 | $\mathbf{8.41}_{\pm 0.05}$ | $\mathbf{7.87}_{\pm 0.05}$ | $\mathbf{6.50}_{\pm 0.01}$ | 100 | $8.22_{\pm 0.10}$ | $\mathbf{7.68}_{\pm 0.03}$ | $\mathbf{6.50}_{\pm 0.09}$ |

We conducted ablation studies on two key parameters for the distillation of D-ODE solvers: the scale $S$ and the batch size $|B|$. The scale $S$ determines the number of sampling steps for the teacher, with the teacher going through $S$ times more denoising steps compared to the student. A larger scale $S$ results in a better target generated by the teacher and can be viewed as increasing the guidance strength of the teacher during distillation. It is also crucial to choose an appropriate batch size $|B|$ since the optimal $\lambda$ is estimated on a single batch $B$ and then reused for other batches. Thus, the batch size should be large enough to encompass different modes of samples within the dataset, while excessively large batch size may not fit into GPU memory.

We tested various scales in Table 3a using noise prediction models trained on CIFAR-10. As the scale increases, the FID score consistently improves across different NFE values. With a larger scale $S$, the student is strongly guided by the teacher's accurate target, resulting in a lower FID. However, the effect of the guidance scale weakens with increasing NFE. This is reasonable since the performance of student sampling depends heavily on that of teacher sampling, and the teacher's FID score eventually converges to a certain value. As the maximum number of timesteps is 1000 for discrete timesteps, scales 20 and 30 at 50 NFE generate samples guided by the same teacher sampling.

Regarding Table 3b, D-ODE solvers with various batch sizes also exhibit clear differences. As the batch size increases, both the FID score and variance tend to decrease. With relatively large NFE values, FID scores and variance converge to a certain point. As the effect of distillation diminishes with higher NFE, even a small batch size results in low variance. We choose a batch size of 100 for most datasets, which is sufficient to capture the inherent variety of the dataset and reduce variance compared to a smaller batch size.

# G MORE COMPARISON

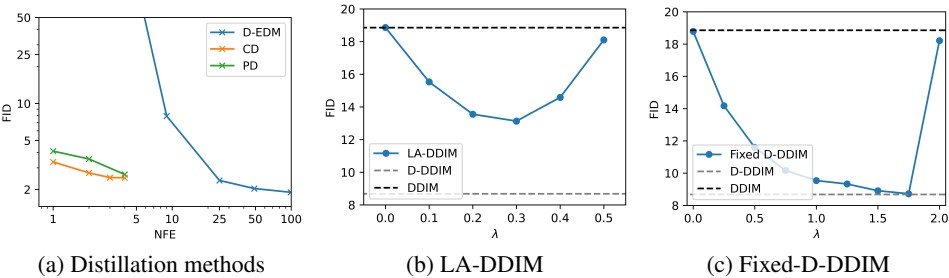

(a) Distillation methods  (b) LA-DDIM  (c) Fixed-D-DDIM

Figure 7: (a) FID scores over NFE for distillation methods (CD, PD, and D-EDM). (b) FID scores over $\lambda$ with LA-DDIM. (c) FID scores over $\lambda$ with Fixed-D-DDIM

Table 4: Comparison with learning-free samplers on CIFAR-10 with noise prediction models. The best FID is bolded.

| NFE | 10 | 25 | 50 |
|---|---|---|---|
| DDIM | 18.85 | 9.79 | 7.17 |
| D-DDIM | **8.67** | 8.18 | 6.55 |
| Fixed-D-DDIM ($\lambda = 0.5$) | 11.45 | **7.00** | **5.27** |
| LA-DDIM ($\lambda = 0.1$) | 15.24 | 8.57 | 6.29 |

In this section, we present further comparisons between D-ODE solvers, previous learning-based methods (knowledge distillation), and learning-free methods. Figure 7a displays FID scores with varying NFE on CIFAR-10. It includes consistency distillation (CD) (Song et al., 2023), which can perform one-step or few-step sampling, and progressive distillation (PD) (Salimans & Ho, 2021), allowing sampling with steps in a geometric sequence (e.g., 1, 2, 4, ..., 1024). On the other hand, D-EDM requires at least two steps to utilize previous denoising outputs.

Overall, CD outperforms other methods in terms of FID on one-step generation. However, it is important to note that this comparison does not account for training time. For instance, Song et al. (2023) reported that consistency models on CIFAR-10 utilized 8 Nvidia A100 GPUs for training. On the other hand, simply generating 50K samples for 30 steps takes less than 30 minutes on a single A100 GPU, achieving similar sample quality to consistency models. while CD and PD are attractive options for practitioners with ample computational resources, given their ability to enable one-step generation, the major advantage of D-ODE solvers lies in their capacity to enhance existing ODE solver-based samplers with minimal modifications and fast optimization.

Recently, Zhang et al. (2023) introduced lookahead diffusion models which enhance the FID scores of existing ODE solvers by refining mean estimation using previous data predictions. They achieve this by extrapolating previous predictions of initial data to approximate the target data. Unlike D-ODE solvers, lookahead models require parameter $\lambda$ to be chosen through grid search, with a default setting of $\lambda = 0.1$ during experiments. Following their configuration, we compare lookahead diffusion models of DDIM, so-called LA-DDIM, with our D-DDIM in Table 4. The table shows that, except at 50 NFE, D-DDIM outperforms LA-DDIM.

Inspired by LA-DDIM, we also experiment with fixing $\lambda_t$ in D-DDIM as a constant $\lambda$ and optimizing it through grid search. We refer to this modified approach as Fixed-D-DDIM. In Figure 7b and 7c, we conduct grid searches on $\lambda$ using a 10-step sampler on CIFAR-10. Additionally, we provide the FID scores of DDIM and D-DDIM as reference points (dotted lines). Despite the grid search performed on LA-DDIM, it is unable to match the FID of D-DDIM. Notably, Fixed-D-DDIM achieves the same FID as D-DDIM with sufficient grid search. This suggests that leveraging denoising outputs is a more efficient strategy than relying on initial data predictions. Moreover, Fixed-D-DDIM further

improves upon D-DDIM's performance at 25 and 50 NFE, indicating the potential for finding an even better $\lambda$ value that results in a lower FID. Future research directions could explore various methods to efficiently determine $\lambda$. It is important to highlight that the FID of LA-DDIM and Fixed-D-DDIM varies depending on the chosen $\lambda$. However, D-DDIM's advantage over other methods is its independence from grid search, with sampling times comparable to DDIM.

## H MORE EXPERIMENT AND ANALYSIS FIGURES

We present extra experiment results in Figure 8 with noise prediction models on CIFAR-10, CelebA64, and ImageNet128. In Figure 9, more analysis figures like Figure 5 are shown with different pixels. We also show more qualitative results in Figure 10, 11, 12, 13, and 14.

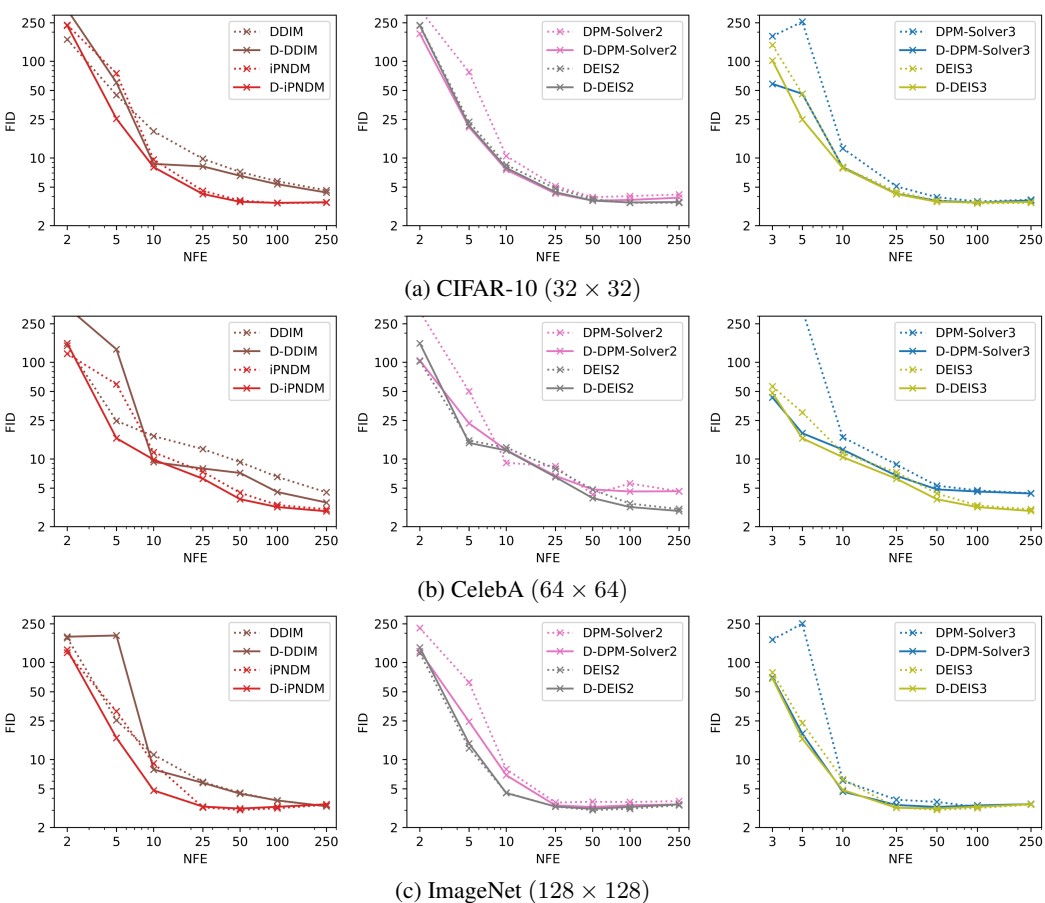

Figure 8: Image quality measured by FID ↓ with varying NFE {2, 5, 10, 25, 50, 100, 250}. For DPM-Solver3 and DEIS3, we use 3 NFE instead of 2 NFE as the third-order method requires at least three denoising outputs. Dotted lines denote ODE solvers (DDIM, iPNDM, DPM-Solver, and DEIS) while straight lines represent the applications of D-ODE solver to them (D-DDIM, D-iPNDM, D-DPM-Solver, and D-DEIS).

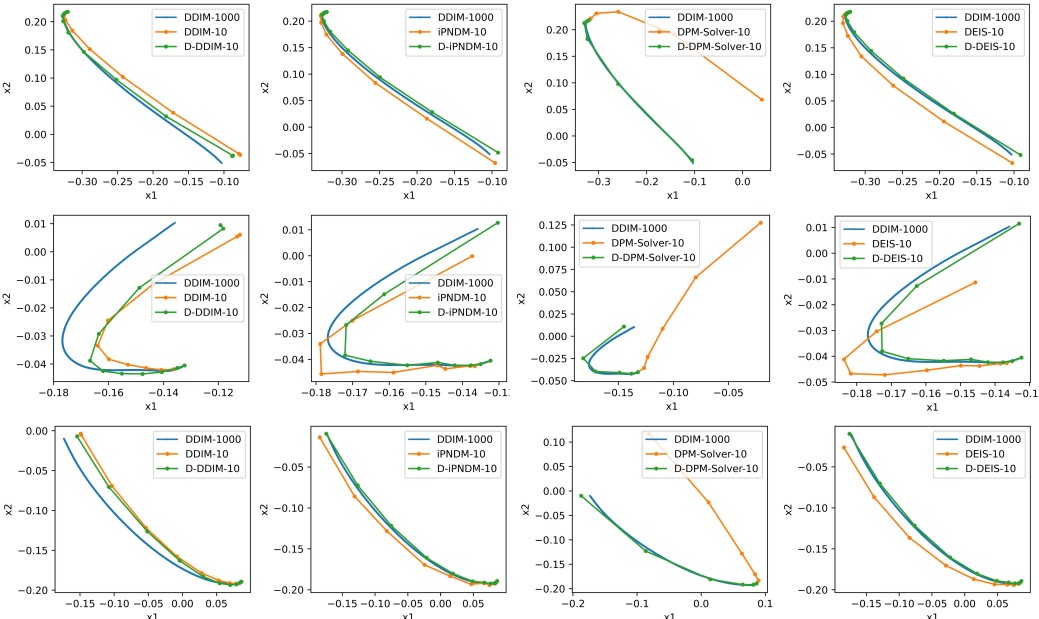

Figure 9: Update path of randomly selected two pixels in the images. The result of 1000-step DDIM is used as our target. These figures are drawn with 1000 samples using a noise prediction model trained on CIFAR-10.

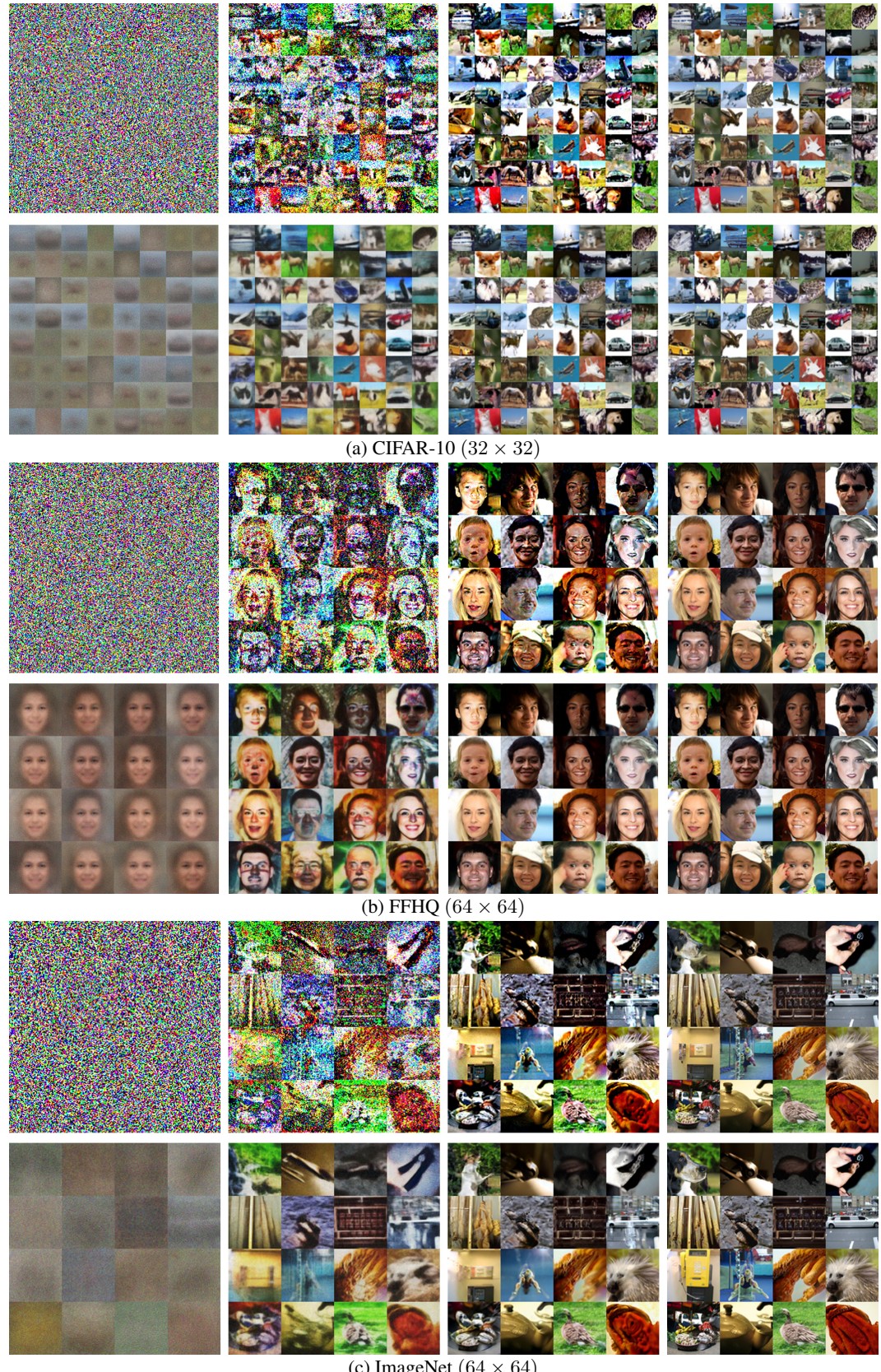

(a) CIFAR-10 ($32 \times 32$)

(b) FFHQ ($64 \times 64$)

(c) ImageNet ($64 \times 64$)

Figure 10: Qualitative results of CIFAR-10 ($32 \times 32$), FFHQ ($64 \times 64$), and ImageNet ($64 \times 64$) with data prediction models. We compare EDM (top) and D-EDM (bottom) in each subfigure with NFE $\in \{3, 5, 9, 25\}$.

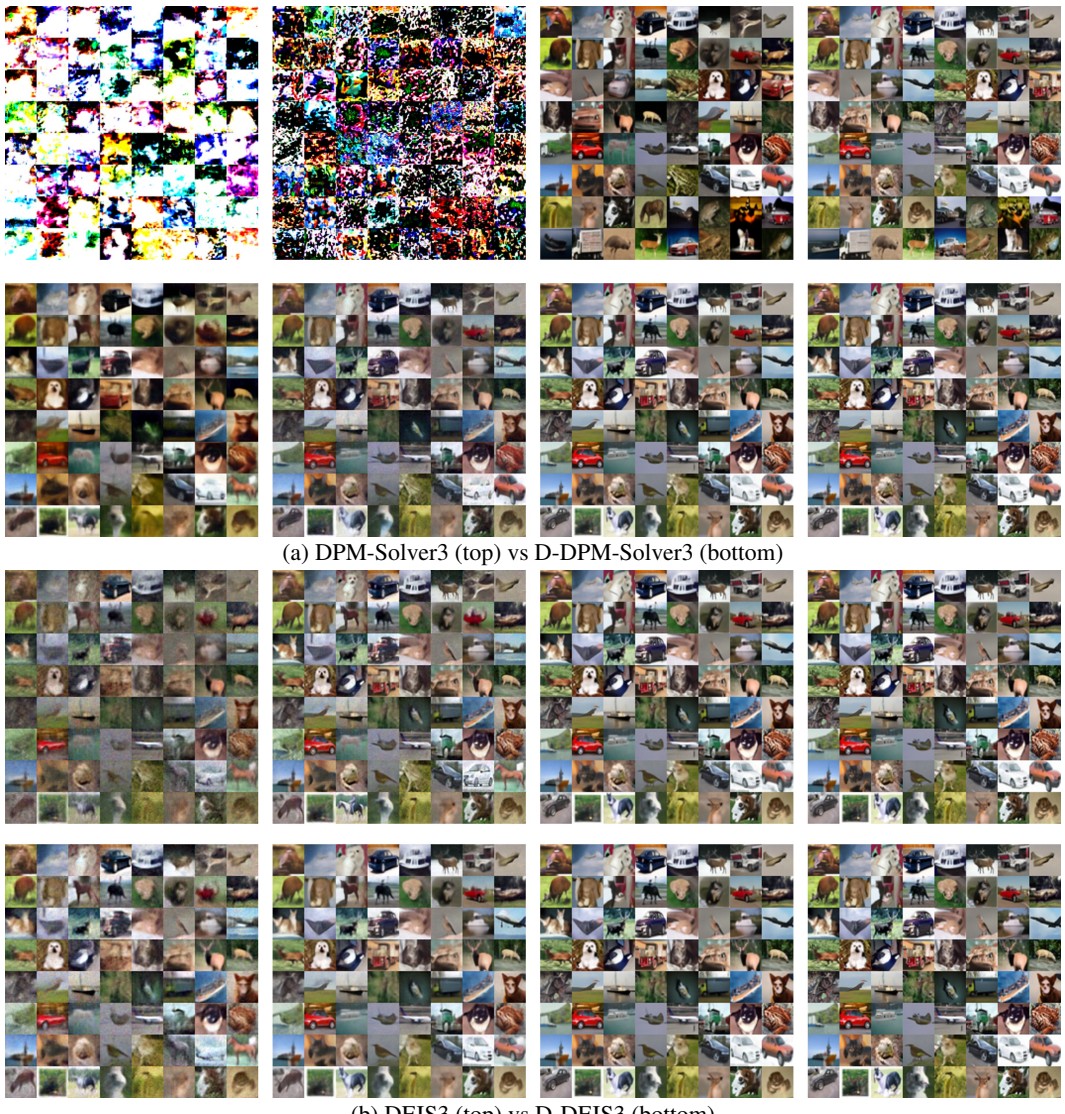

(a) DPM-Solver3 (top) vs D-DPM-Solver3 (bottom)

(b) DEIS3 (top) vs D-DEIS3 (bottom)

Figure 11: Qualitative results of CIFAR-10 ($32 \times 32$) with noise prediction models. We compare ODE-solvers (DPM-Solver3, DEIS3) and D-ODE solvers (D-DPM-Solver3, D-DEIS3) in each sub-figure with NFE $\in \{3, 5, 10, 25\}$.

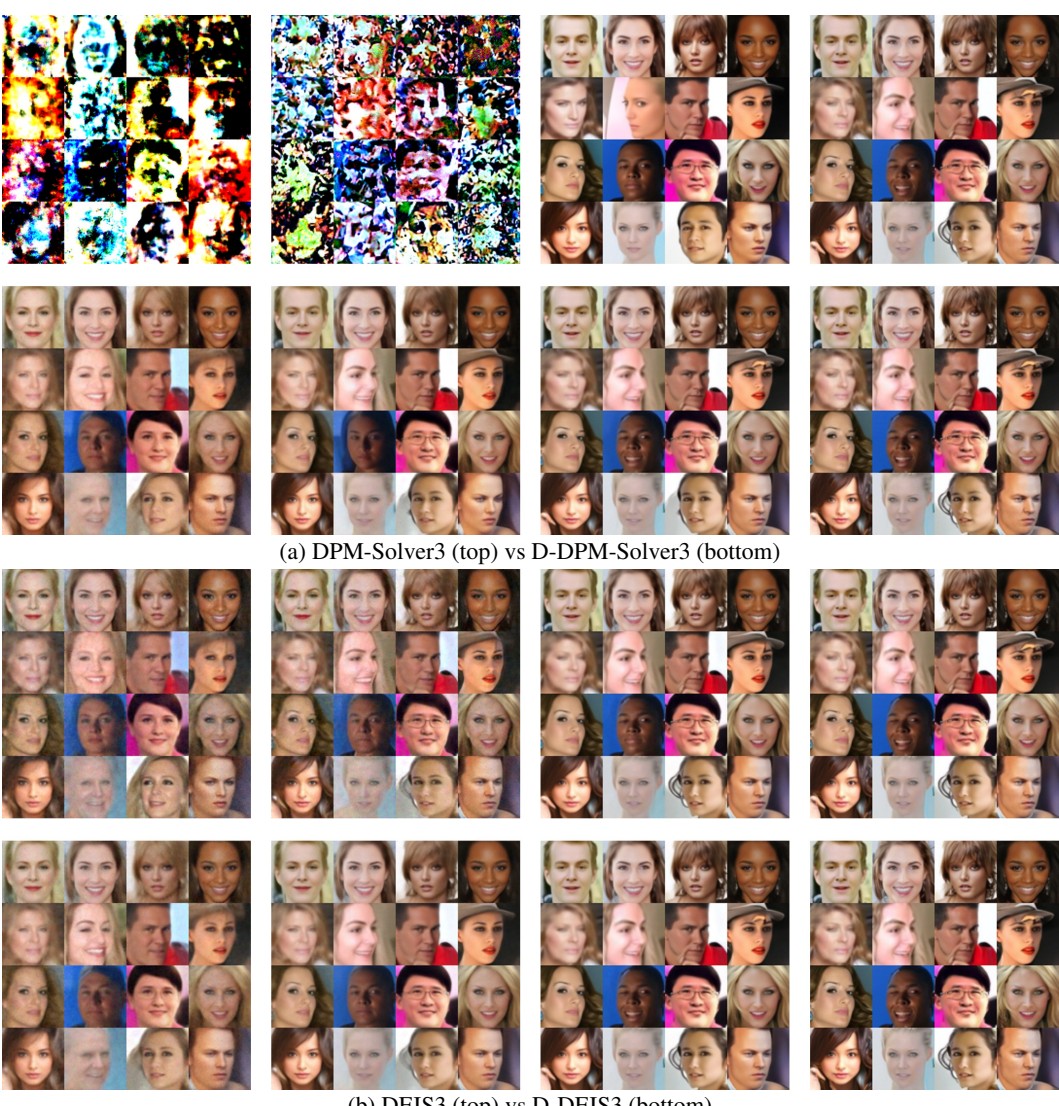

(a) DPM-Solver3 (top) vs D-DPM-Solver3 (bottom)

(b) DEIS3 (top) vs D-DEIS3 (bottom)

Figure 12: Qualitative results of CelebA $(64 \times 64)$ with noise prediction models. We compare ODE-solvers (DPM-Solver3, DEIS3) and D-ODE solvers (D-DPM-Solver3, D-DEIS3) in each subfigure with NFE $\in \{3, 5, 10, 25\}$.

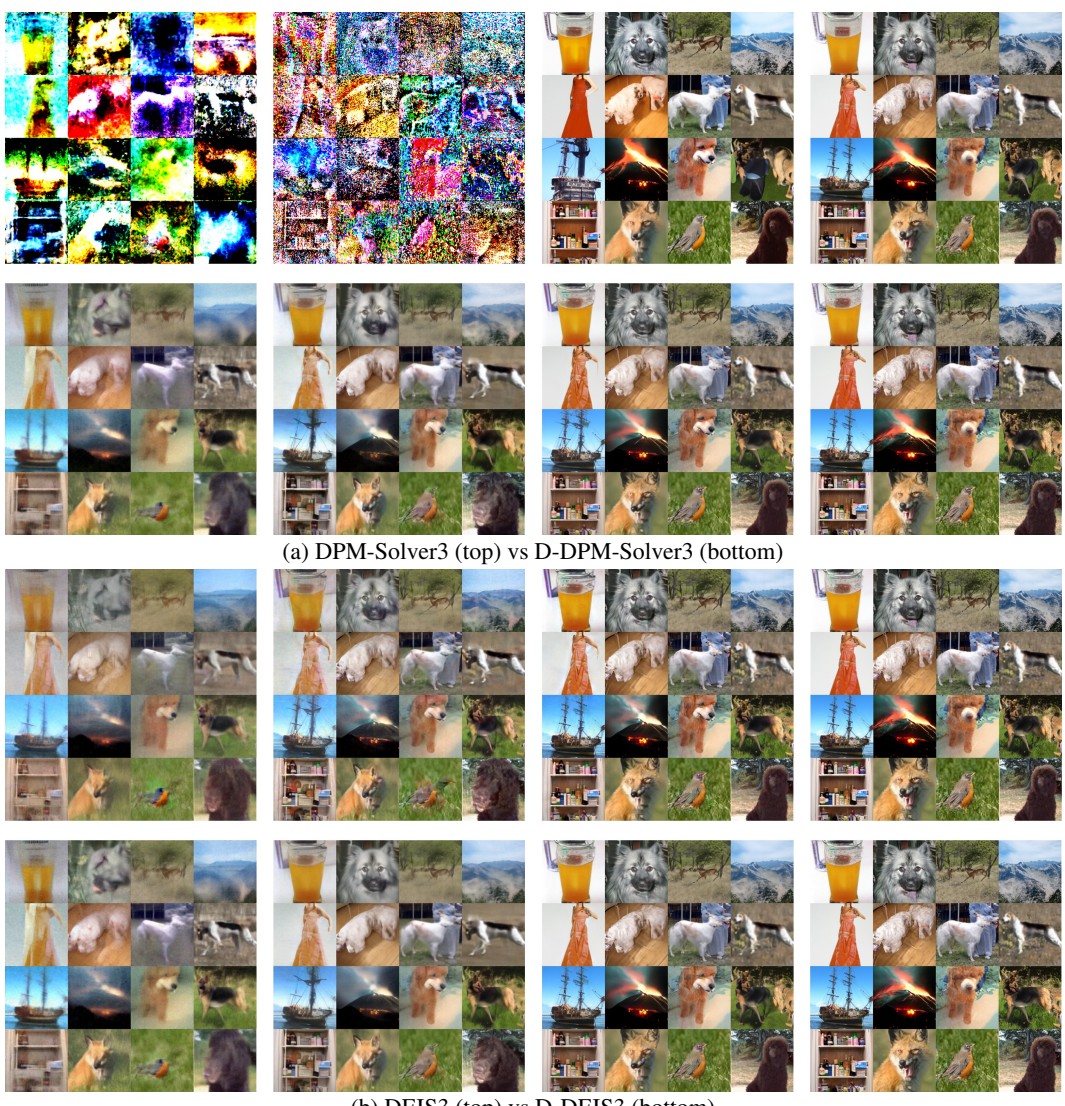

(a) DPM-Solver3 (top) vs D-DPM-Solver3 (bottom)

(b) DEIS3 (top) vs D-DEIS3 (bottom)

Figure 13: Qualitative results of ImageNet ($128 \times 128$) with noise prediction models. We compare ODE-solvers (DPM-Solver3, DEIS3) and D-ODE solvers (D-DPM-Solver3, D-DEIS3) in each subfigure with NFE $\in \{3, 5, 10, 25\}$.

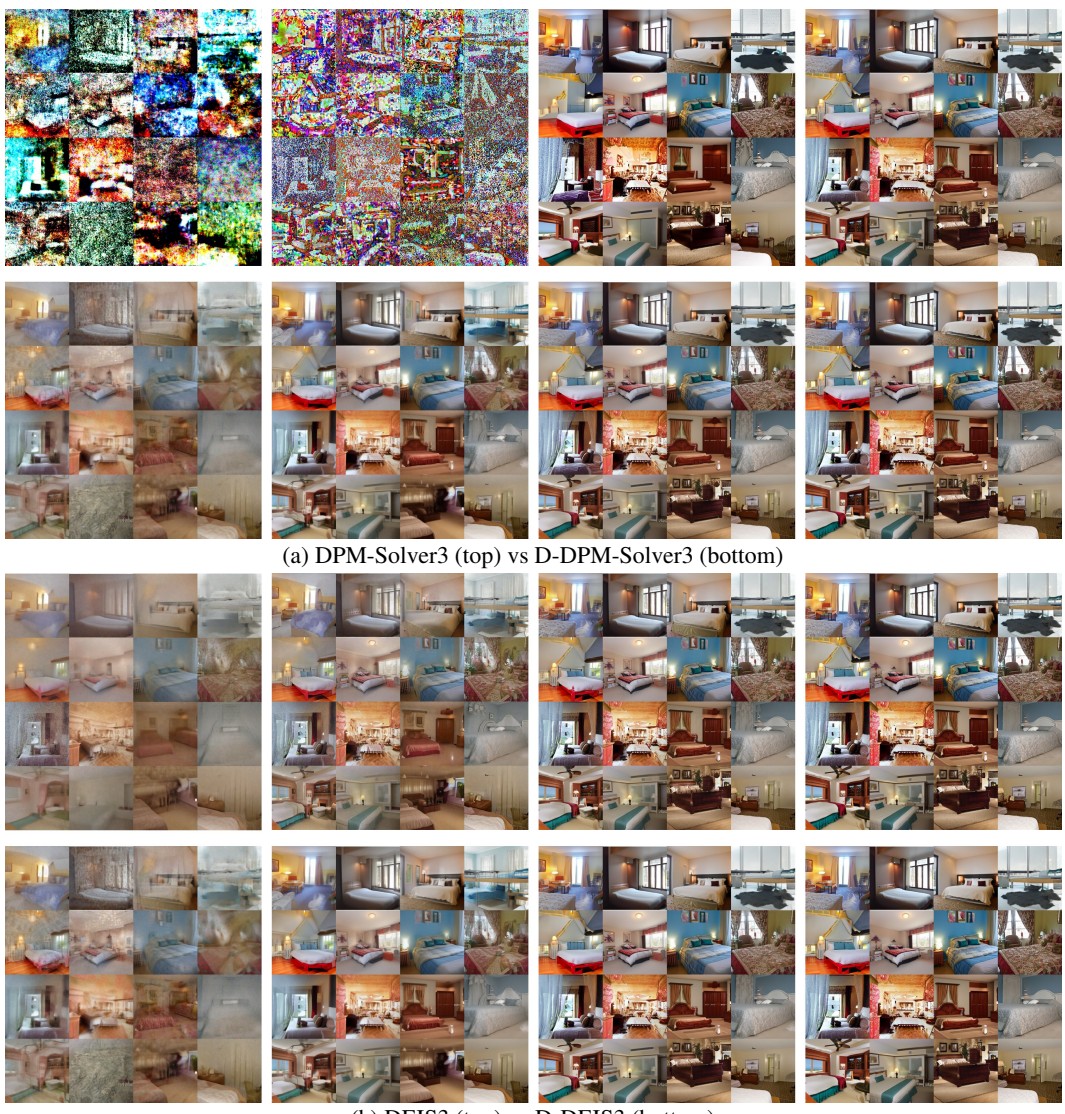

(a) DPM-Solver3 (top) vs D-DPM-Solver3 (bottom)

(b) DEIS3 (top) vs D-DEIS3 (bottom)

Figure 14: Qualitative results of LSUN Bedroom ($256 \times 256$) with noise prediction models. We compare ODE-solvers (DPM-Solver3, DEIS3) and D-ODE solvers (D-DPM-Solver3, D-DEIS3) in each subfigure with NFE $\in \{3, 5, 10, 25\}$.

