# OpenReview forum: "Distilling ODE Solvers of Diffusion Models into Smaller Steps"
_ICLR.cc/2024/Conference — ICLR 2024 Conference Withdrawn Submission_

### Official Review · Reviewer_931n · 2023-10-24

**Soundness:** 3 good
**Presentation:** 3 good
**Contribution:** 1 poor
**Rating:** 5
**Confidence:** 4

**Summary:**

This paper considers accelerating the sampling processes of a number of existing ODE solvers (including both 1st and high order ODE solvers) for small NFEs. The basic idea is to optimize a scalar parameter lambda_i in front of a gradient vector per timestep t_i for approximating the ODE integration over [t_i, t_{i-1}]. The optimization for lambda_i is performed by solving a quadratic optimization problem which measures the difference between the estimated diffusion state x_{i-1} obtained via the student ODE solver and that obtained by the teacher ODE solver with much larger number of timesteps.  By doing so, the integration approximation accuracy for the student ODE solver should be improved. Experiments on FID evaluation confirm the effectiveness of the new method.

**Strengths:**

Different from progressive distillation which needs to fine-tune the student model, this paper considers optimizing the scalar lambda_i in front of some gradient vector per diffusion timestep. The gradient vector is a function of estimated clean images. The optimization process is very cheap since it only involves quadratic optimization. The resulting FID improvement is reasonable.

**Weaknesses:**

My main concern is the novelty of this paper. An arXiv paper released in April this year has a similar research idea as this paper. The paper title is "On Accelerating Diffusion-Based Sampling Process via Improved Integration Approximation". The arXiv paper considered improving the sampling performance of EDM, DDIM, DPM-Solver, and SPNDM for small NFEs.  One main difference between the two papers is that the arXiv paper considered optimizing a number of coefficients in front of some gradient vectors per diffusion timestep rather than one coefficient as being optimized in this paper. Furthermore, the gradient vectors considered in the arXiv paper can be functions of estimated Gaussian noises or estimated clean data.

It seems that the authors of this paper is not aware of the arXiv paper at all.

**Questions:**

It is interesting to know the performance of the new method for advanced applications like text-to-image generation.

---

### Official Review · Reviewer_Wdr7 · 2023-10-30

**Soundness:** 2 fair
**Presentation:** 3 good
**Contribution:** 2 fair
**Rating:** 5
**Confidence:** 4

**Summary:**

This paper proposes a straightforward distillation approach that optimizes the ODE solvers rather than the denoising networks. To capture the teacher's trajectory, this paper adds a learnable parameter to ODE solvers. Then the optimization procedure is based on a batch of samples of the trajectories of teacher samplers.

**Strengths:**

$\cdot$ The writing of this paper is clear and self-contained, which helps the reviewer to quickly follow.

$\cdot$ Rather than distilling the denoising network, this paper proposes a new method to distill the ODE solver.

$\cdot$ The experiment results show improvement over the baseline methods.

**Weaknesses:**

$\cdot$ The improvement over the baseline method is limited as results shown in Figure 3 and Figure 4, especially for the DEIS3 ODE solver. The improvement on DDIM and EDM Heun are better, but these two ODE solvers do not perform well when NFEs are small. Although the proposed method requires less computational time compared with methods like Consistency Distillation or Progressive Distillation, the weak performance hinders its application in practice.

$\cdot$ Seems to miss quantitative evaluation (e.g. FID, CLIP score on stable diffusion) for text-to-image tasks.

**Questions:**

See weaknesses.

---

### Official Review · Reviewer_pnZP · 2023-10-31

**Soundness:** 2 fair
**Presentation:** 1 poor
**Contribution:** 2 fair
**Rating:** 3
**Confidence:** 4

**Summary:**

This work proposes D-ODE solver, a new type of diffusion ODE solver based on modified denoising networks. Different from pevious solvers with are based on either noise-pred or data-pred networks, this work uses a new denoising prediction by linearly combining some "denoising outputs" in two adjacent time steps and then train the linear coefficients to design samplers. Experiments show that the proposed method can be used for improving various of samplers, including DDIM, DPM-Solver, DEIS, PNDM and EDM.

**Strengths:**

- The proposed method is parallel with other well-deigned fast samplers and can be used for further improving them.
- The experiments show the effectiveness of the proposed method.

**Weaknesses:**

- Major:
  - The writing needs to be greatly improved, and it is rather hard to follow the paper by only reading the main text. For example, the **$D_t$ and $D_\theta$ are not defined in the main text**, so that I cannot understand the main method until I carefully read the appedix. Please add a rigorous defination of the "denoising prediction".
  - Lack of a detailed discussion with multi-step ODE solvers, such as DEIS[1] and DPM-Solver++[2]. In fact, the proposed linear combination of current step and the previous step is exactly the core idea of the multi-step solvers. If we choose $D_\theta$ and $D_t$ as the original network (i.e., noise-pred for DEIS or data-pred for DPM-Solver++), the proposed method is almost equivalent to them, except for the requirements for estimating the linear coefficients. Besides, the main advantage of multi-step solvers is that they are more robust and have less discretization errors when using fewer steps, that's why DEIS and DPM-Solver++ outperform other single-step solvers. It seems that the main improvement in this work is exactly the multi-step paradigm.
  - The D-DEIS in the appendix seems to be very tricky, because it uses the combination of the "linear combination of noise-pred network", not the combination of the noise-pred network. Why use this setting? Is there any principle for choosing $D_t$ and $D_\theta$?
  - In Sec.4.2, please also compare with DPM-Solver++ because it is based on data-prediction model.
- Minor:
  - The writing for a vector-valued variable or function should be consistent. For example, please both use the bold format for the noise variable and the noise-pred network in Eq(6), and similarly for the data-pred network in Eq(7).



[1] Zhang, Qinsheng, and Yongxin Chen. "Fast sampling of diffusion models with exponential integrator." *arXiv preprint arXiv:2204.13902* (2022).

[2] Lu, Cheng, et al. "Dpm-solver++: Fast solver for guided sampling of diffusion probabilistic models." *arXiv preprint arXiv:2211.01095* (2022).

**Questions:**

1. Please clarify the rigorous definition of the "denoising prediction".

2. Please discuss with multi-step solvers in details, including DEIS and DPM-Solver++, and compare with them in experiments.

3. Why DEIS use the combination of noise-pred networks for the "denoising prediction", but not use the noise prediction network itself? Are there any differences? In my opinion, "the combination of noise-pred networks" have larger errors because, for example, if we use DEIS-2 for time t, t-1, then the combination involves time t, t-1, t-2, which corresponds to DEIS-3. What is the difference?